# Pharmacological perturbation of the phase-separating protein SMNDC1

Lennart Enders [1], Marton Siklos[1], Jan Borggräfe[2,3], Stefan Gaussmann [2,3], Anna Koren[1], Monika Malik[1], Tatjana Tomek[1], Michael Schuster [1], Jiří Reiniš[1], Elisa Hahn [1], Andrea Rukavina[1], Andreas Reicher[1], Tamara Casteels[1,5], Christoph Bock [1,4], Georg E. Winter [1], J. Thomas Hannich[1], Michael Sattler [2,3] & Stefan Kubicek [1] ✉

SMNDC1 is a Tudor domain protein that recognizes di-methylated arginines and controls gene expression as an essential splicing factor. Here, we study the specific contributions of the SMNDC1 Tudor domain to protein-protein interactions, subcellular localization, and molecular function. To perturb the protein function in cells, we develop small molecule inhibitors targeting the dimethylarginine binding pocket of the SMNDC1 Tudor domain. We find that SMNDC1 localizes to phase-separated membraneless organelles that partially overlap with nuclear speckles. This condensation behavior is driven by the unstructured C-terminal region of SMNDC1, depends on RNA interaction and can be recapitulated in vitro. Inhibitors of the protein's Tudor domain drastically alter protein-protein interactions and subcellular localization, causing splicing changes for SMNDC1-dependent genes. These compounds will enable further pharmacological studies on the role of SMNDC1 in the regulation of nuclear condensates, gene regulation and cell identity.

Survival motor neuron domain-containing protein 1 (SMNDC1), also called Survival of motor neuron-related-splicing factor 30 (SPF30), is an essential splicing factor required for the formation of the spliceosome[1,2]. To promote spliceosome assembly SMNDC1 binds to methylated arginines on Sm-proteins using its Tudor domain[2,3], similar to its better-studied paralog survival of motor neuron (SMN) protein[4–6]. The Tudor domain structures of both proteins are highly conserved, revealing binding of their substrate symmetrically di-methylated arginine (sDMA) in an aromatic cage through cation-π interactions[7]. Functionally, both proteins play essential and apparently opposite roles in the regulation of gene expression and cell identity in the endocrine pancreas. Patients and animal models with SMN mutations experience increased numbers of glucagon producing alpha cells and a reduction of insulin producing beta cells[8]. In contrast, for

SMNDC1 we recently showed that its knock-down causes the upregulation of insulin in α-cells through splicing changes in key chromatin remodelers and induction of the beta cell transcription factor PDX1[9]. SMNDC1 further is essential for cell proliferation in different contexts, and a recent study reported worse survival in hepatocellular carcinoma patients with high SMNDC1[10]. SMNDC1 knock-down led to decreased proliferation and migration of hepatocellular carcinoma cells, establishing SMNDC1 as a potential therapeutic target.

Both SMN and SMNDC1 show distinct and focal subcellular localization patterns. The SMN Tudor domain is sufficient for formation of a phase-separated compartment dependent on the dimethylarginine (DMA) modification of binding proteins[11] and was shown to be required for the regulation of the phase-separated stress granules via symmetric dimethylarginine (sDMA)[12]. Arginine methylation in RGG/

[1]CeMM Research Center for Molecular Medicine of the Austrian Academy of Sciences, Lazarettgasse 14, 1090 Vienna, Austria. [2]Helmholtz Munich, Molecular Targets and Therapeutics Center, Institute of Structural Biology, Neuherberg 85764 München, Germany. [3]Technical University of Munich, TUM School of Natural Sciences, Department of Bioscience, Bavarian NMR Center, Garching 85748 München, Germany. [4]Medical University of Vienna, Institute of Artificial Intelligence, Center for Medical Data Science, Währinger Straße 25a, 1090 Vienna, Austria. [5]Present address: Sloan Kettering Institute, 1275 York Avenue, New York, NY 10065, USA. ✉e-mail: skubicek@cemm.oeaw.ac.at

RG motifs recognized by Tudor domains can affect phase separation of Fused in sarcoma (FUS)[13,14] and other proteins[15], and further Tudor domain containing proteins themselves have been shown to be involved in phase separation[16,17].

SMNDC1 has a speckled localization within the nucleus that – based on co-localization – was attributed to the sub-nuclear structures Cajal bodies and nuclear speckles[2], which were later defined as prime examples of membraneless organelles[18], i.e. biomolecular condensates formed by liquid-liquid phase separation (LLPS). These assemblies can consist of proteins, nucleic acids, and other molecules and are found both in the cytoplasm and the nucleus[19,20]. An important feature present in many proteins that were found to undergo LLPS are intrinsically disordered regions (IDRs), which do not adopt a well-defined globular structure. IDRs can enable multiple and multivalent interactions that mediate binding to other proteins[21]. Many RNA-binding proteins (RBPs), including SMNDC1, were found to phase separate together with RNA, but also with chromatin[22]. Amongst other factors phase separation behavior can be initiated by RNA[23] and regulated by the secondary structure of RNAs and the ratio of RNA to RBPs[24–26]. Given the fact that the nucleus and its sub-compartments are enriched in IDR-containing proteins (IDPs)[27] and the obvious abundance of negatively charged nucleic acids (both DNA and RNA) the nucleus is primed for LLPS[28]. Functionally, these LLPS events control gene expression within the different nuclear compartments[29] from the formation of heterochromatin[30,31] over transcription by RNA polymerase II[32] to RNA processing and (alternative) splicing[33].

Tudor domains have not been targeted extensively by small-molecule inhibitors. Only recently, a study disclosed a fragment unspecifically binding to both SMN and SMNDC1 in isothermal titration calorimetry (ITC), and with cellular specificity for SMN[34]. Similarly, specific agents perturbing biomolecular condensation events are lacking, and pharmacological approaches often rely on unspecific agents like 1,6-hexanediol[35] at concentrations of several hundred millimolar.

Here, we study the phase-separating behavior of SMNDC1 both in vitro and within cells and we develop specific inhibitors against its Tudor domain influencing the sub-cellular localization and phase separation of their target.

## Results

### SMNDC1 co-localizes with nuclear speckle markers

To identify features associated with subcellular SMNDC1 localization, we analyzed the protein sequence by comparing predictions for disordered regions by MetaDisorder[36] and for the full-length structure by AlphaFold[37,38] (Fig. 1a). The experimentally solved Tudor domain structure[7] (residues 64-128) and two interacting N-terminal alpha-helices (residues 2-25, and 30-52) are visible both in the AlphaFold prediction and in the disorder tendency plot as ordered regions. AlphaFold in addition predicts a long C-terminal alpha-helix, for which however currently no other experimental evidence exists.

We employed an endogenous tagging system that targets introns and introduces a GFP-tag as an artificial exon[39] to characterize SMNDC1's cellular functions. To rule out disrupting effects of the tag on protein localization, we targeted all of SMNDC1's introns in murine alphaTC1 cells, and then isolated clonal sublines. The targeted introns result in GFP integrations covering all regions of the protein, including one at the N-terminus (before residue 1), the N-terminal region (residue 40), the Tudor domain (residue 88), and a long stretch in the C-terminal region (residue 142, residue 193) which is predicted to be disordered[36] (Fig. 1a, b). Furthermore, we also tagged intron 2–3 in human HAP1 cells. Typically, these monoallelic tagging events resulted in cells expressing both un-tagged and GFP-tagged SMNDC1 at comparable levels as shown by western blot (WB) (Fig. 1c, quantifications and full membranes Supplementary Fig. 1b, c). The GFP-tag within the

Tudor domain (intron 3-4) showed the lowest relative expression levels, indicating possible interference with folding efficiency.

AlphaFold structure predictions[37,38] for SMNDC1 with and without GFP in the different introns revealed that the GFP-tag does not seem to disrupt the overall structure of the protein (Supplementary Fig. 1a). All structural elements such as the N- and C-terminal α-helices and the Tudor domain (red) are predicted to form normally, even when the GFP-tag interrupts the Tudor domain (intron 3-4). Accordingly, all of the different intron tagged clones, including the intra-Tudor GFP integration showed consistent subcellular localization patterns (Fig. 1b). These GFP fusions showed the same speckled nuclear localization avoiding DNA-dense regions as observed for the endogenous protein by antibody-based immunofluorescence (IF) (Fig. 1d). During M-phase of the cell cycle SMNDC1 dissipated to the whole cell and formed distinct droplets called mitotic interchromatin granules[40,41] (Fig. 1e), a behavior which is typical for nuclear speckle proteins[42,43]. SMNDC1 also reacted to the overexpression of the cell-cycle dependent kinases DYRK3 and CLK1, which is known to dissolve nuclear speckles[43,44], with a loss of its focal nuclear localization (Supplementary Fig. 1d).

To further characterize SMNDC1's localization in the nucleus, we co-stained cells with antibodies against SMNDC1 and SC35, a marker for nuclear speckles. Both signals overlap to a large degree and avoid chromatin-dense regions, whereby SMNDC1 shows a wider, less focal distribution (Fig. 1f, co-localization analysis Supplementary Fig. 1e). To be able to visualize nuclear speckles in live cells we RFP-tagged SRRM2 in the SMNDC1-GFP-tagged cells (Fig. 1g). SRRM2 is the target of the SC35 antibody[45] and scaffolding protein of nuclear speckles[46].

Endogenously tagged SMNDC1-GFP and SRRM2-RFP co-localized to a large degree, both in interphase and during mitosis (Fig. 1h). Even though co-localization was maintained in the mitotic interchromatin granules, there SMNDC1 showed a higher degree of diffuse localization, leading to a lower average Pearson correlation score compared to interphase cells (Supplementary Fig. 1f). Overall, we find that SMNDC1 shows behavior and localization typical for proteins in nuclear speckles, which have been described as membraneless organelles in the nucleus formed by LLPS.

### SMNDC1 undergoes biomolecular condensation in vitro and in cellular systems

A common way to prove phase-separating behavior of a protein is to show its ability to form droplets in a purified form in vitro. To do so, we expressed and purified full SMNDC1 with an N-terminal GFP-tag and mixed it with PEG-8000 as a surrogate for the crowded environment of a cell. We observed droplet formation (Fig. 2a) and fusion of droplets (Fig. 2b). Subsequently we tested the influence of other biomolecules and salt concentration on droplet formation (Fig. 2c). Addition of RNA to the PEG-8000 containing buffer enhanced SMNDC1's droplet formation while high NaCl concentrations prevented droplet formation. Digestion of RNA by RNase led to the dissolution of droplets, even after their formation (Fig. 2d). RNA also physically localized to the protein droplets (Fig. 2e).

To further understand which part of the protein is responsible for the formation of droplets, we fused different SMNDC1 truncations (Fig. 1a) to GFP and subjected them to the same treatment in buffer containing RNA and PEG-8000. These experiments clearly displayed that the C-terminal region after the Tudor domain (constructs 5 and 6), which is predicted to be intrinsically disordered[36], was sufficient to induce droplet formation with RNA (Fig. 2f, see Fig. 1a for a scheme of the truncated forms), which fit the predicted IDR scores[36] (Fig. 1a). We also confirmed that the Tudor domain alone (construct 3) cannot form droplets, consistent with previous literature[11].

To show the reversibility of phase separation in vivo, the aliphatic alcohol 1,6-hexanediol which interferes with weak hydrophobic

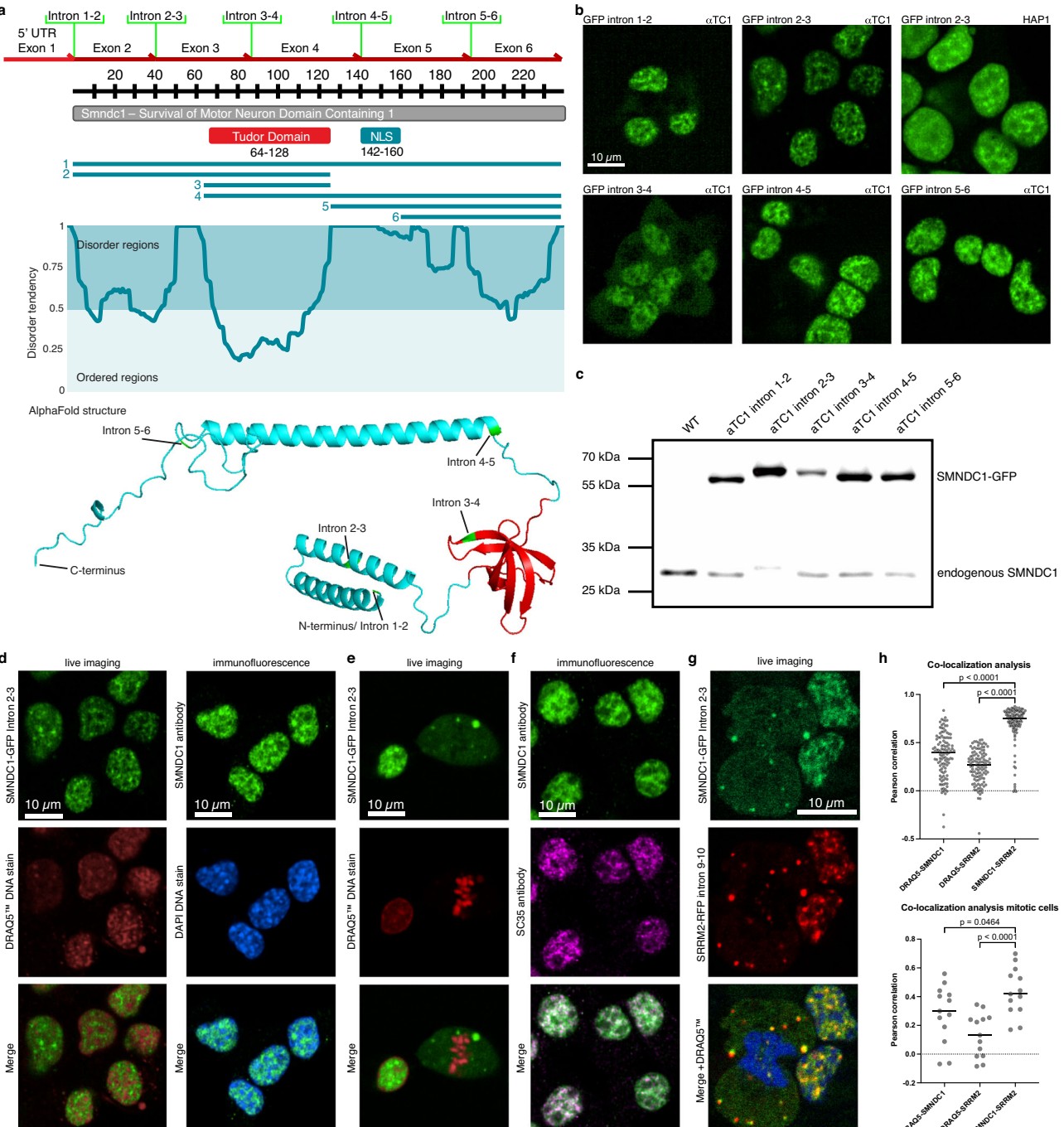

**Fig. 1 | SMNDC1 co-localizes with nuclear speckle markers. a** Overview of SMNDC1's structure with numbered truncations (for Fig. 2f), intrinsic disorder prediction plot (MetaDisorder[36]), AlphaFold structure prediction with Tudor domain marked in red and positions of GFP intron-tags in green. **b** Live images of clonal cell lines (αTC1 and HAP1) with the endogenous GFP-tag in green. **c** Immunoblots showing expression of WT SMNDC1 and SMNDC1-GFP fusion proteins in clonal cell lines with GFP-tag in different introns. **d** Live (SMNDC1-GFP intron 2-3, αTC1) and immunofluorescence images (αTC1 WT) with nuclear staining (DRAQ5™ in red/ DAPI in blue). **e** Live imaging (SMNDC1-GFP intron 2-3, αTC1) with

DRAQ5™ nuclear staining showing a cell during M-phase. **f** Immunofluorescence images (αTC1 WT) with SMNDC1 (green) and SC35 antibody (magenta), overlap of green and magenta is white. **g** Live imaging (SMNDC1-GFP intron 2-3, SRRM2-RFP intron 9-10, αTC1) with DRAQ5™ nuclear staining (blue) showing a cell during telophase. **h** Co-localization analyses of interphase (n = 114) and mitotic (n = 13) cells, Pearson correlation between different channels of maximum intensity projections of z-stack images. Data shown as scatter plot + median line, analyzed by two-tailed, unpaired t-test.

interactions is often used to dissolve protein condensates[35]. SMNDC1-GFP exhibited the expected phenotype in live cells treated with 1,6-hexanediol by losing its focal localization within the nucleus (Fig. 2g). Another way to characterize the molecular dynamics and mobility of phase-separating proteins in cells is to analyze the diffusion of a fluorescently labeled protein by fluorescence recovery after

photobleaching (FRAP). When bleaching SMNDC1-GFP and SRRM2-RFP, fluorescence recovered within 30 seconds (Fig. 2h), consistent with liquid-like behavior rather than protein aggregation. These data provide evidence that SMNDC1 undergoes phase separation, both in vitro and in membraneless organelles within the nucleus, presumably nuclear speckles.

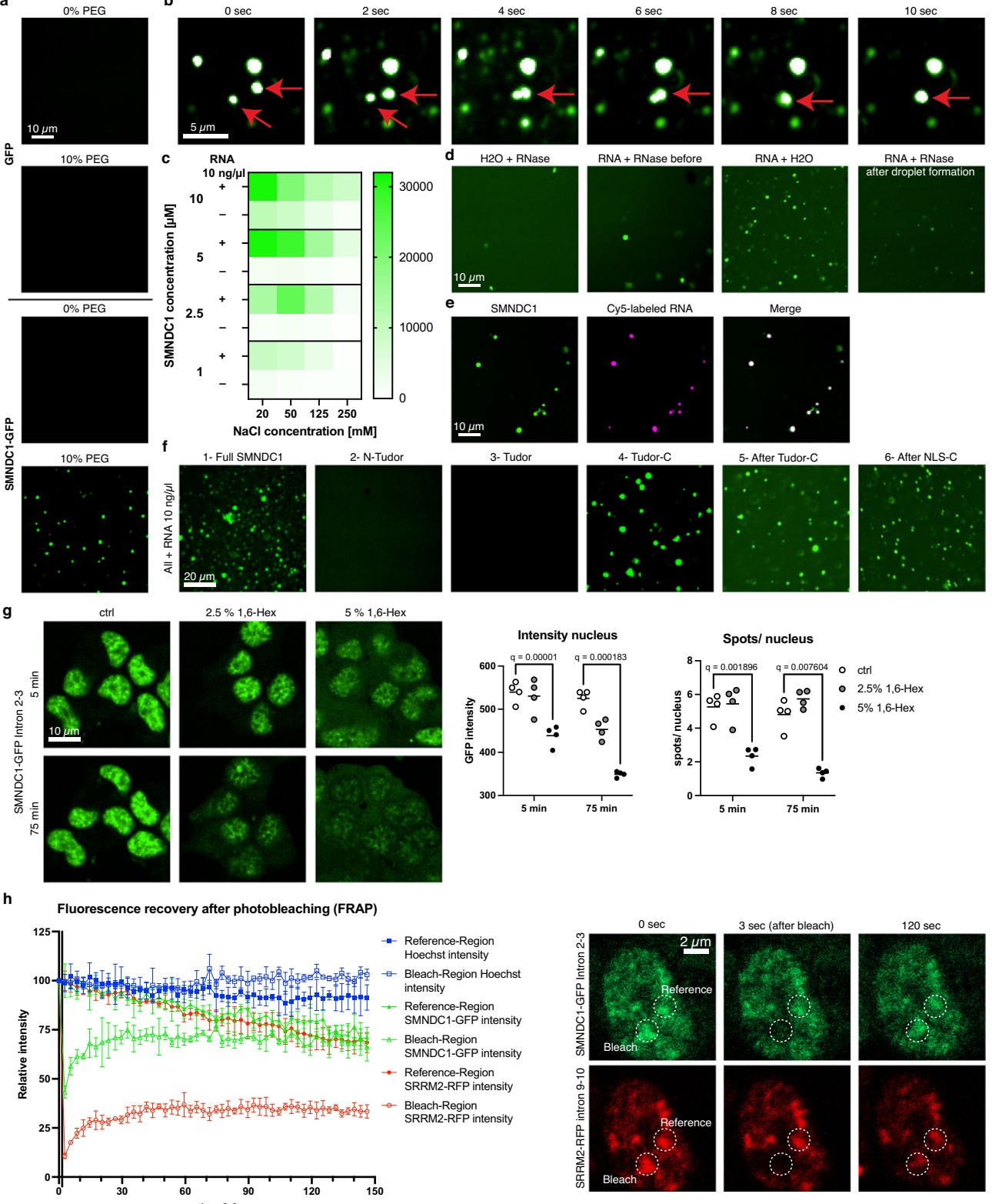

**Full-length SMNDC1 interacts with nuclear speckle proteins**

We set out to characterize SMNDC1's interactome using proximity labeling by overexpressing an SMNDC1-APEX2 fusion protein (Fig. 3a). Compared to classical co-immunoprecipitation (Co-IP), this recently developed method[47] is better suited to capture weak and transient interactions as they are expected in phase-separated compartments like nuclear speckles. In addition to full-length SMNDC1 (APEX2-

SMNDC1[FL]), we also performed proximity labeling with a fusion protein of APEX2 with a truncated SMNDC1 consisting of only the Tudor domain and therefore lacking N-terminal and C-terminal regions, and the nuclear localization signal (NLS) (APEX2-SMNDC1[TD]) (Supplementary Fig. 2a). To verify our approach, we performed proximity labeling followed by IF staining against SMNDC1 and biotin. APEX2-SMNDC1[FL] caused biotinylation in the areas where SMNDC1 is localized: nuclear

**Fig. 2 | SMNDC1 shows biomolecular condensation in vitro and in cellular systems. a** In vitro droplet formation assay with 10 μM GFP or SMNDC1-GFP fusion protein +/− 10% PEG-8000. **b** In vitro droplet formation assay of SMNDC1-GFP over time with droplet fusion event, marked by red arrows. **c** In vitro droplet formation assay of SMNDC1-GFP with quantified number of droplets with different protein and NaCl concentrations, +/− 10 ng/μl RNA. **d** In vitro droplet formation assay of SMNDC1-GFP with the addition of 10 ng/μl total cellular RNA and RNase. **e** In vitro droplet formation assay of SMNDC1-GFP (green) with 100 ng/μl Cy5-labeled RNA (magenta), overlap white. **f** In vitro droplet formation assay of different truncations of SMNDC1-GFP + 10 ng/μl total cellular RNA. **g** Live imaging (SMNDC1-GFP intron 2-3, αTC1), cells were treated with 2.5% or 5% 1,6-hexanediol. Quantifications of GFP

intensity and GFP spots/nucleus in different clonal cell lines. Data presented as scatter plot with mean line ($n = 4$), analyzed by two-tailed, multiple paired $t$-tests with False Discovery Rate q calculated by Two-stage step-up[90]. **h** Fluorescence recovery after photobleaching (FRAP) experiment in SMNDC1-GFP intron 2-3, SRRM2-RFP intron 9-10, αTC1-cells. Left: Relative intensity of Hoechst (blue), SMNDC1-GFP (green), and SRRM2-RFP (red) in reference (filled symbols) and bleach region (empty symbols) over time. Data plotted as mean with standard deviation, $n = 3$. Right: representative images of nucleus with marked reference and bleach region at 3 different timepoints, 0 s (before bleaching), 3 s (directly after bleaching), and 120 s (after recovery).

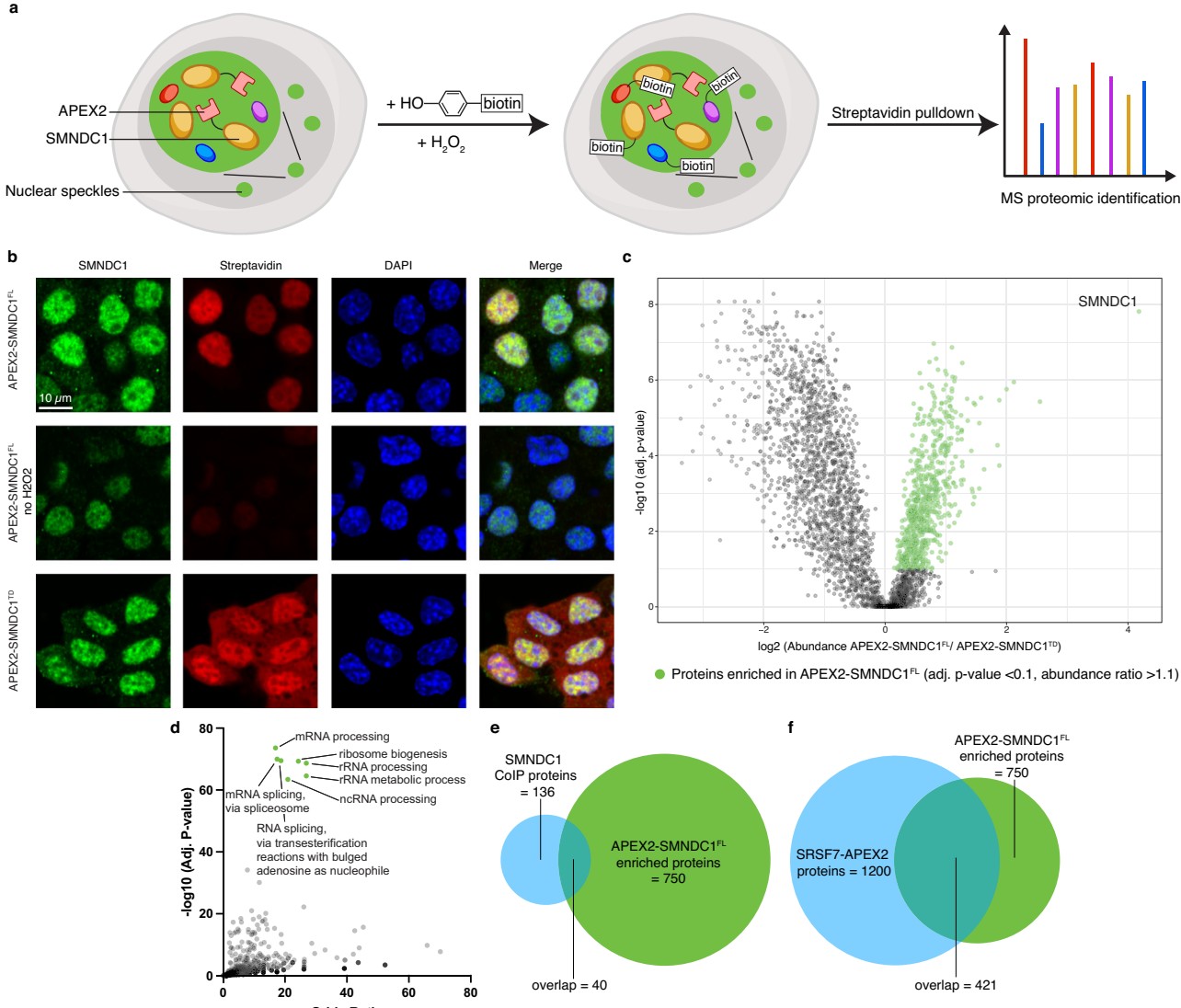

**Fig. 3 | Characterization of SMNDC1's interactome by proximity labeling.**
**a** Scheme of proximity labeling by APEX2 fusion proteins followed by mass spectrometry-based proteomics. **b** Immunofluorescence images (αTC1 WT) with staining against SMNDC1 (green), Biotin via Streptavidin (red) and nuclear staining DAPI (blue). **c** Volcano plot showing log2 abundance against -log10 adjusted $p$-value (one-way ANOVA, Benjamini-Hochberg correction for multiple comparisons) of APEX2-SMNDC1$^{FL}$ versus APEX2-SMNDC1$^{TD}$ biotinylated and enriched proteins. Highlighted dots in green indicate 750 enriched proteins (adjusted $p$-value < 0.1,

abundance ratio > 1.1). **d** Enrichr analysis of APEX2-SMNDC1$^{FL}$ enriched proteins, GO Biological Process 2021 terms plotted with their odds ratio and their adjusted $p$-value (Benjamini–Hochberg method for correction for multiple hypotheses testing). Terms with a -log10 adjusted $p$-value >60 colored green. **e** Venn diagram showing the overlap of proteins identified by SMNDC1-CoIP (light blue), and APEX2-SMNDC1$^{FL}$ enriched (green). **f** Venn diagram showing the overlap of proteins identified by SRSF7-APEX2 (light blue), and APEX2-SMNDC1$^{FL}$ enriched (green).

while avoiding chromatin-dense regions (Fig. 3b). Much less biotinylation was observed when omitting $H_2O_2$. The control overexpression of APEX2-SMNDC1$^{TD}$ on the other hand showed a uniform localization throughout the cell and a corresponding biotinylation pattern. On a

western blot, a ladder of biotin-labeled proteins was visible, but absent when leaving out the $H_2O_2$ during the labeling. More proteins appear to be labeled by the ubiquitously localized APEX2-SMNDC1$^{TD}$ fusion (Supplementary Fig. 2b).

Analyzing the biotinylated and enriched proteins by mass spectrometry (MS), we identified and quantified a large number of proteins (~3200) in the proximity of APEX2-SMNDC1[FL] and APEX2-SMNDC1[TD]. Compared to the proximity interactome of APEX2-SMNDC1[TD], APEX2-SMNDC1[FL] showed overall less interactions (Fig. 3c). We attribute this to the higher specificity of interactions happening with the correctly localized full form of SMNDC1. The fact that SMNDC1 itself was enriched in APEX2-SMNDC1[FL] over APEX2-SMNDC1[TD] suggests that labeling in trans works better if SMNDC1 is correctly localized and concentrated in its phase-separated compartment leading to more SMNDC1 protein in its proximity. Similarly, proteins known to be localized to the nucleus were not depleted in APEX2-SMNDC1[FL] over APEX2-SMNDC1[TD], reflecting the loss of correct localization when the NLS is missing (Supplementary Fig. 2c).

We then filtered for proteins enriched in APEX2-SMNDC1[FL] over APEX2-SMNDC1[TD] (adjusted $p$-value < 0.1, abundance ratio > 1.1) which reduced the number of proteins we considered specific interactors of SMNDC1[FL] to 750. As expected, we found an enrichment of proteins associated with mRNA processing, and more specifically splicing, but also an enrichment of proteins associated with ribosome biogenesis and rRNA processing amongst these (Fig. 3d). When comparing these interactors to an SMNDC1 Co-IP dataset generated in our lab[9] we found a significant overlap but confirm that proximity labeling can detect more and different interactions compared to a Co-IP (Fig. 3e). A majority of APEX2-SMNDC1[FL] interactors was also identified by SRSF7-APEX2 proximity labeling[48] (Fig. 3f), suggesting that APEX2-SMNDC1[FL] proximity labeling did enrich for proteins localized to nuclear speckles. Furthermore, we compared the interactors to proteins identified as symmetrically di-methylated on arginine residues in a deep protein methylation profiling study[49] (Supplementary Fig. 2d). Since these interactions are expected to be mediated through the Tudor domain APEX2-SMNDC1[TD] should bind these proteins, too. Consequently, only a small subset was enriched in APEX2-SMNDC1[FL] over APEX2-SMNDC1[TD]. We therefore also compared the sDMA-modified proteins to all proteins identified in our SMNDC1-APEX2 experiments and found most of them (67 out of 87 known sDMA-modified proteins). There was also an enrichment, although to a lesser degree, of proteins with asymmetrical di-methylations. These protein sets partially overlap, as the same arginine sites can often alternatively be symmetrically or asymmetrically di-methylated.

Overall, we found a large interactome of SMNDC1 enriched for proteins interacting with RNA, localized to nuclear speckles, and with known sDMA modifications. We therefore suspected that the Tudor domain is responsible for a subset of SMNDC1's specific interactions.

## A screen for small molecule SMNDC1 Tudor domain inhibitors

To pharmacologically perturb SMNDC1 function, we set out to identify small molecule inhibitors of SMNDC1's Tudor domain based on perturbing its interaction with a dimethylarginine peptide. To establish an AlphaScreen[50], we coupled donor beads to purified SMNDC1's (or SMN's) Tudor domain via a His-Tag and acceptor beads to a biotinylated peptide corresponding to the C-terminal region of the Small nuclear ribonucleoprotein Sm D3 containing four sDMAs (Fig. 4a). These interaction partners had previously been used in the structural study of SMNDC1 and SMN[7]. Protein domains were purified employing their His-Tag (Supplementary Fig. 3a). To identify ideal concentrations for screening, we performed a cross-titration of Tudor domains and binding peptides (Supplementary Fig. 3b). Since the AlphaScreen signal was sufficient for screening, we reduced the concentration of acceptor and donor beads to 5 µg/ml (Supplementary Fig. 3c).

Using this set-up with our in-house library of ~90,000 compounds (overview over screening strategy Fig. 4b, Supplementary Table 1), we identified 511 hits with signal <50% of control (POC) (Fig. 4c). Since the AlphaScreen is susceptible to unspecific quenching of the singlet oxygen energy transfer we then performed a counter-screen using a

crosslinking peptide which combines both affinity tags and therefore always brings donor and acceptor beads in close proximity (Fig. 4d). This led to a reduction to 40 hits that selectively inhibited the interaction between SMNDC1, and its arginine methylated binding partner. These compounds we next tested in dose response with the Tudor domains of both SMNDC1 and SMN. Several chemical scaffolds (Fig. 4e–k) of inhibitors were discovered by this screen, with IC$_{50}$ values of 0.2 to 2 µM and different degrees of selectivity between the different Tudor domains. The molecules with the best physicochemical and structural properties were then used for the design of further analogs, aiming at improving potency and selectivity (Fig. 5).

## 4-arylthiazole-2-amines show clear structure-activity relationships as SMNDC1 Tudor domain inhibitors

Among the most potent hit compounds were 2-amino-4-arylthiazoles and benzoxazepines. Of these classes, benzoxazepines had undesirable physicochemical properties including very low polarity (Fig. 4f, clogP = 6.33) along with poor solubility. Therefore, we abandoned this series after testing a limited set of analogs (Supplementary Table 2).

We then selected the 4-arylthiazole-2-amine series for thorough exploration of structure-activity relationships, also due to the synthetic ease of access (Fig. 5). We found the 2-pyridyl substitution to be important for binding affinity, as its replacement with other aryl groups led to drastic loss of potency (e.g., compounds 3-7). A 2-substituted pyrrole could be used with some loss of potency in compound 8. Omission of the aromatic group by replacement with ethoxycarbonyl resulted in complete loss of activity (compound 9).

In contrast, a wide variety of substituents were tolerated in the 2-position of the thiazole. Even the unmodified aminothiazole 13 showed a submicromolar IC$_{50}$. This compound also served as the synthetic starting point for this series and related chemical probes. The amide linkage between the thiazole and the aryl group is dispensable for activity as demonstrated by the alkylamine 17 and the sulfonamide 18. Replacement of the aromatic amine by guanidine 19 decreased the IC$_{50}$. The aryl amide could be substituted or replaced with a wide selection of groups, both aromatic and aliphatic rings with minor effects on potency (compounds 21-24).

Among the most potent compounds were compound 1 and its morpholinosulfamoyl analog 2. The arylsulfonamide could be replaced with other groups with minimal loss of potency (compounds 25-28), whereby larger substituents as in compounds 25 and 28 increased selectivity for SMNDC1 over SMN.

The five-membered heterocycle in the core scaffold could be replaced with the isomeric scaffold 2-(pyridin-2-yl)thiazol-4-amine in compound 14. The third possible isomer, compound 15, had significant loss of activity and preferentially inhibited SMN over SMNDC1. When the thiazole was replaced with an analogous oxazole in the compound 16, there was a 40-fold drop in potency. Replacement of the thiazole with 1,2,4-thiadiazoles resulted in inactive compounds. Substitution of the 5-position of the thiazole of 1 with a methyl group was tolerated without loss of potency (compound 10) but an ethyl group or a bromine atom decreased the IC$_{50}$ threefold (compounds 11 and 12).

These extensive structure-activity relationships revealed features that are absolutely essential for the binding of this scaffold to Tudor domains and indicate for substructures required for achieving selectivity between SMNDC1 and SMN. In the following biological characterization, we focus on compound 1 as a potent Tudor domain inhibitor, validate findings with the SMNDC1-specific compound 28 and use the inactive compound 9 as a negative control.

## 2-amino-4-arylthiazoles bind the methyl-arginine pocket of the SMNDC1 Tudor domain

To prove specific binding of SMNDC1 inhibitors to the aromatic cage of SMNDC1 and to obtain structural information on the binding modes,

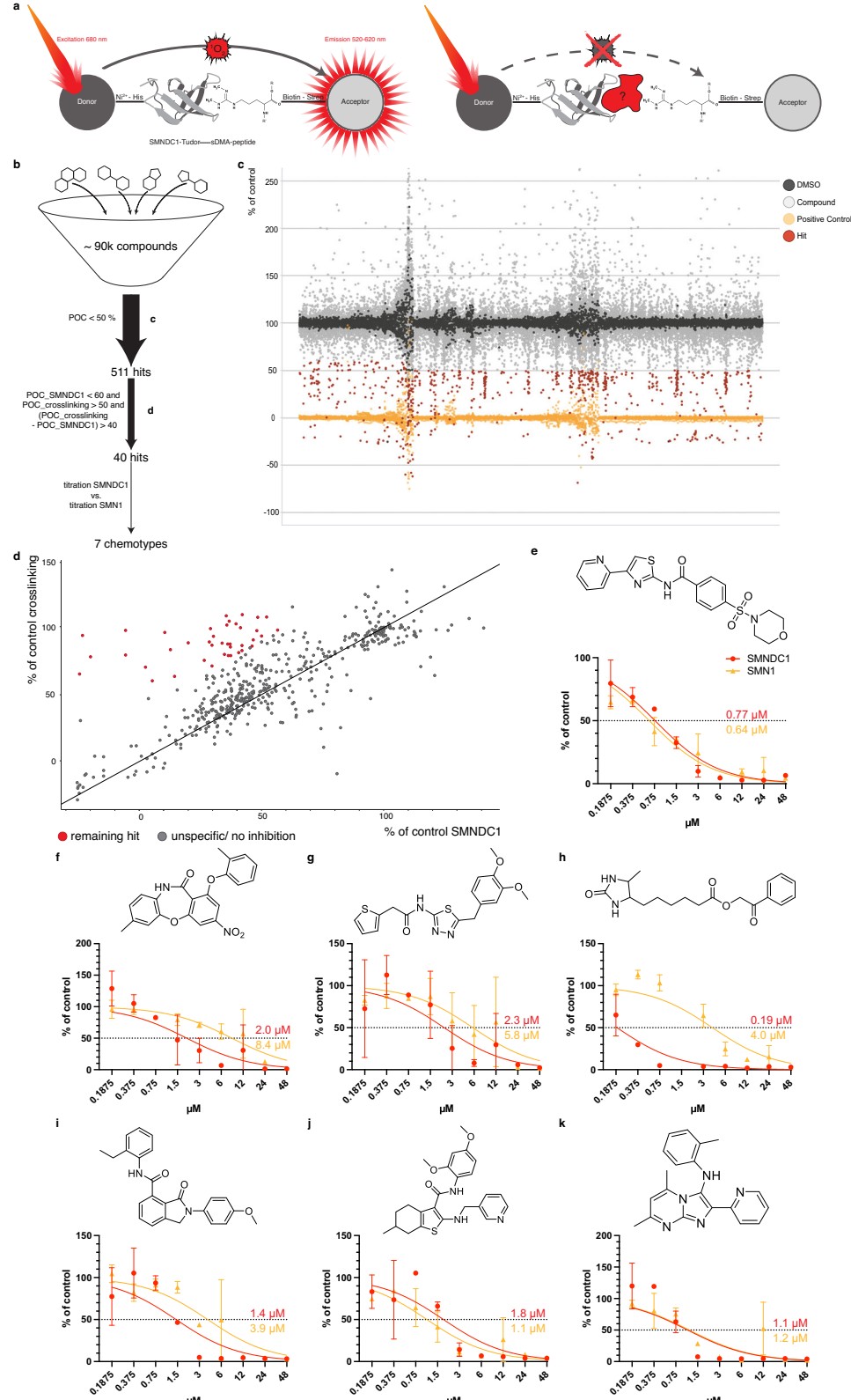

**Fig. 4 | Identification of an inhibitor against SMNDC1's Tudor domain. a** Scheme of AlphaScreen set-up with the NMR-structure of SMNDC1's Tudor domain (PDB: 4A4H)[7]. **b** Screening strategy starting with ~90,000 compound library. **c** Overview of AlphaScreen of full ~90,000 compound library (light gray) with DMSO (dark gray), positive control (quencher, yellow), and compound hits (red). **d** AlphaScreen percentage of DMSO control with SMNDC1/ sDMA-peptide vs crosslinking peptide. Remaining hits marked in red. **e–k** Chemical structure and AlphaScreen 9-point compound titration with SMNDC1/ sDMA-peptide (red) vs. SMN/ sDMA-peptide (yellow). Data presented as mean +/− SD (*n* = 2).

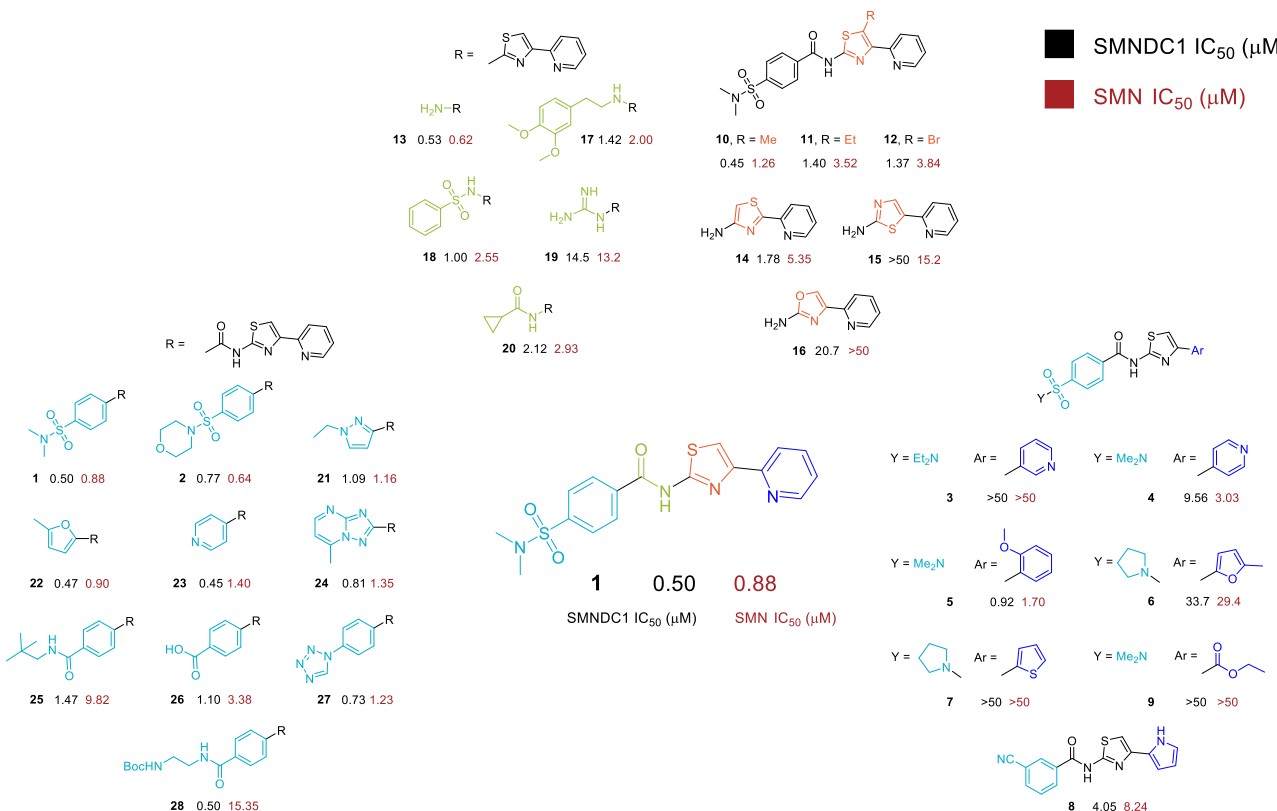

**Fig. 5 | Structure-activity relationships of the 2-aryl-4-aminothiazole Tudor domain inhibitors.** Color-coded chemical structures illustrating structure activity relationships for compound **1**. Modifications to the 2-aryl moiety are shown in dark blue, replacements of the thiazole group in orange, linker modifications in green and arylamide analogs in turquoise. Underneath are IC$_{50}$ values (μM) for SMNDC1 in black and SMN in dark red.

we applied nuclear magnetic resonance spectroscopy (NMR). The Tudor domains of SMN (residues 84-147) and SMNDC1 (residues 65-128) were expressed in isotope-enriched medium and purified as described elsewhere[7]. Since organic solvents such as dimethyl sulfoxide showed non-specific binding to the Tudor domains and led to interference in the NMR experiments, we attempted to use an aqueous, buffered solution of compound **1**, which however exhibited insufficient solubility for NMR experiments. We therefore prepared an aqueous, buffered solution of the monobasic phosphate salt of compound **13** and assessed its concentration by comparing signal intensities to a DSS standard. NMR titrations of compound **13** with SMN$_{84-147}$ and SMNDC1$_{65-128}$ showed significant chemical shift perturbations (CSP) with binding kinetics reflecting fast-exchange (gradual change of chemical shift with increasing ligand concentration[51]) for both proteins (Fig. 6a, c). CSP are highly sensitive to changes of the local chemical environment of the observed nuclear spin and therefore excellent reporters to map binding sites of a ligand and (potentially associated) conformational changes. The largest CSP are observed for the amino acids forming the aromatic cage (W83, Y90, F108, Y111) and the surrounding residues. Additionally, some parts of the β$_2$-strand show significant chemical shifts with increasing concentration of compound **13**. The affected residues and CSP match very well the ones published for sDMA binding[7], with exception of residues W83 and S84, suggesting a different interaction with the aromatic cage's tryptophan, as well as N113.

In order to obtain higher resolution structural information of the recognition of compound **13** by the Tudor domain we recorded $^{13}$C-filtered NOESY experiments using a 1 mM $^{15}$N,$^{13}$C-labeled SMNDC1 Tudor domain with a 20-fold excess of compound **13**. We could observe a number of contacts between the Tudor domain and the ligand by intermolecular nuclear Overhauser effects (NOE), most prominently with aromatic protons of the Tudor binding site identified by the CSP (Fig. 6b, d, Supplementary Table 3). Using the intermolecular NOEs we calculated a rigid model docking calculation using HADDOCK[52,53], which yielded one cluster with low structural deviation (Supplementary Table 4). The structure indicates that the ring nitrogens of compound **13** are in a *cis* conformation in the complex. The pyridine moiety stacks inside the aromatic cage and its aromatic protons (H2-H5) are showing multiple contacts with the protein's aromatic residues, while the thiazole proton has considerably less contacts to the protein (Fig. 6d, Supplementary Table 3, Supplementary Fig. 4a). The pyridine of compound **13** forms tight π-π stacking contacts with the aromatic rings of F83 and Y111 with distances of 3.7 Å to each, which underlines the importance of an aromatic substituent at the thiazole 4-position. The aromatic moieties of Y90 and F108 stand perpendicular to the pyridine ring while the sidechain of N113 is enclosing it from the opposite site. Overall, the structure shows high similarity to SMNDC1/sDMA (PDB: 4A4H) (Supplementary Fig. 4c) and is fully consistent with the predicted binding mode.

## SMNDC1 Tudor domain inhibitors impact protein localization and splicing

We then went on to analyze the effects of the identified small molecule binders on SMNDC1's phase separation. Using the endogenously tagged cell lines, we observed strong effects on the levels and distribution of SMNDC1. Treating the cells with 50 μM of compound **1** for 12–16 h leads to a loss of SMNDC1 within the nucleus (Fig. 7a, quantification Fig. 7b,). Additionally, the subnuclear distribution changed and less spots were detected within the nucleus (quantification Fig. 7c). These effects were not observed with the negative control compound **9** which lacks the 2-pyridyl crucial for the binding to SMNDC1. Co-staining nuclei with Hoechst showed that nuclear structure was not affected and that these cells were in interphase (Supplementary Fig. 5a). Longer treatment with compound **1** resulted in cell death. While the percentage of AnnexinV

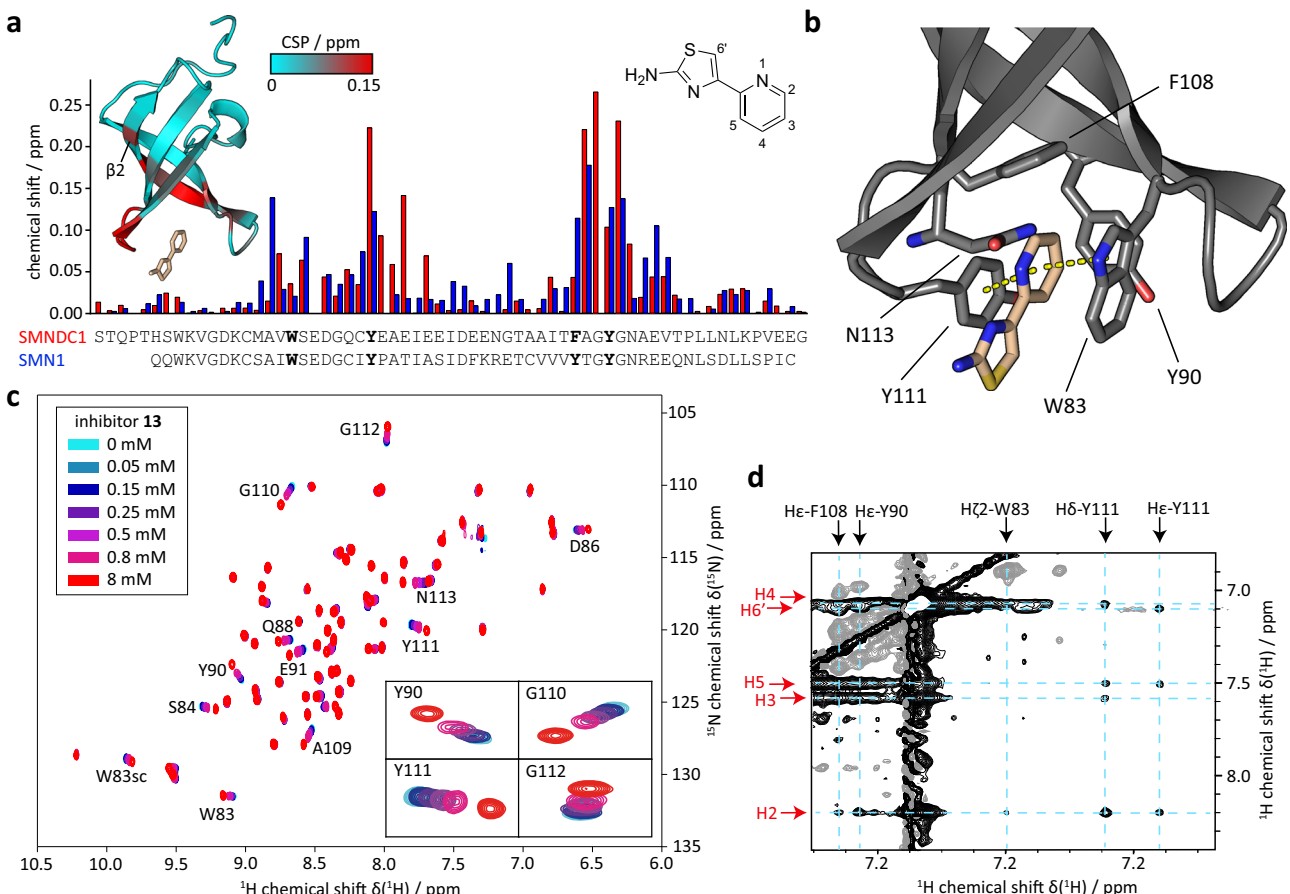

**Fig. 6 | Inhibitor binds at the aromatic cage of the Tudor domain. a** Chemical shift perturbations (CSP) of Tudor domains SMNDC1 (red) and SMN (blue) in presence of 0.8 mM compound **13**. Residues forming the aromatic cage are high-lighted in bold. Left inlet shows cartoon representation of SMNDC1 in complex with compound **13** (light-brown sticks) calculated using semi-rigid body docking of compound **13** based on 23 intermolecular NOE restraints (see **d** and Supplementary Table 3). CSP per residue in presence of 8 mM compound **13** are displayed in cyan to red shades. Right inlet shows compound **13** with numbers indicating the assignment used in d and Supplementary Table 3. **b** Zoomed view of the binding

site with residues forming the aromatic cage shown as sticks. Stacking contacts between the thiazole moiety of compound **13** and tryptophan 83 or tyrosine 111 indicated with dashed lines. **c** Overlay of ¹H,¹⁵N-HSQC spectra of SMNDC1 in the presence of 0, 0.05, 0.15, 0.25, 0.5, 0.8, and 8 mM compound **13**. Zoomed view shows residues in and near the aromatic cages. **d** Section of an in $\omega_1$-¹³C-filtered NOESY spectrum shows crosspeaks between inhibitor protons (red arrows) and aromatic cage protons (black) arrows. The crossing points of the dashed cyan lines indicate the locations of the intermolecular NOEs.

and Propidium Iodide (PI) positive cells was elevated, the majority of cells was not apoptotic or undergoing other forms of cell death at this timepoint (Supplementary Fig. 5b), and cell death was only observed at later timepoints.

Using the cell-line in which SMNDC1 and SRRM2 are both tagged we examined the effects of compound **1** on nuclear speckles. Upon treatment with inhibitor **1**, but not compound **9**, SRRM2 and therefore general organization of nuclear speckles was also affected. The overall SRRM2 intensity upon treatment was slightly reduced, and spots appeared to dissolve into the nucleoplasm (Fig. 7d, quantifications Fig. 7e and Supplementary Fig. 5c). Treating several independent SRRM2-RFP clones replicated the results for inhibitor **1** (Supplementary Fig. 5f). These results could also be confirmed using antibodies against SMNDC1 and SC35 in IF (Supplementary Fig. 5f).

To check whether inhibition of SMNDC1 is indeed responsible for the effects on nuclear speckles, we silenced SMNDC1 with and without the inhibitor (Fig. 7f, images Supplementary Fig. 5g). The knock-down of SMNDC1 also led to a reduction of SRRM2 intensity and even more pronounced to a reduction of SRRM2 spots in the nucleus, confirming the importance of SMNDC1 for the integrity of nuclear speckles. Treatment with the inhibitor could not further increase these effects, hinting that it is not an unspecific effect of the inhibitor that causes the disruption of nuclear speckles. Furthermore, we tested the SMNDC1-

selective compound **28** for its effects on SMNDC1 and SRRM2 locali-zation and could confirm the effects observed for the non-selective compound **1** (Supplementary Fig. 5h), even at lower concentrations (Supplementary Fig. 5i).

To directly test the effect on SMN with its similar Tudor domain, we created cell lines in which SMN1 was endogenously tagged with RFP. Treating these cells with 50 μM of compound **1** for 16 h showed effects on SMN. Overall intensity of SMN decreased while number of spots (supposedly stress granules) in the cytoplasm increased (Sup-plementary Fig. 5j).

Next, we analyzed the effect of compound **1** on the proximity interactome of SMNDC1. Overall, we observed that upon inhibitor treatment, more proteins showed a reduced interaction (volcano plot skewed towards down-regulated side, many more significantly down-regulated than up-regulated proteins, Fig. 7g). This indicates that the inhibitor blocks SMNDC1's function to bind to its interaction partners. Compared to APEX2-SMNDC1^FL, the inhibitor effects in APEX2-SMNDC1^TD were diminished, presumably due to less specific interac-tions in the truncated form at baseline (Supplementary Fig. 6a). 126 proteins were significantly depleted in APEX2-SMNDC1^FL treated with inhibitor vs. none in APEX2-SMNDC1^TD (adjusted *p*-value < 0.05, log2 fold-change ≤−2). Proteins with known sDMA modifications identified in SMNDC1's interactome were among the most depleted

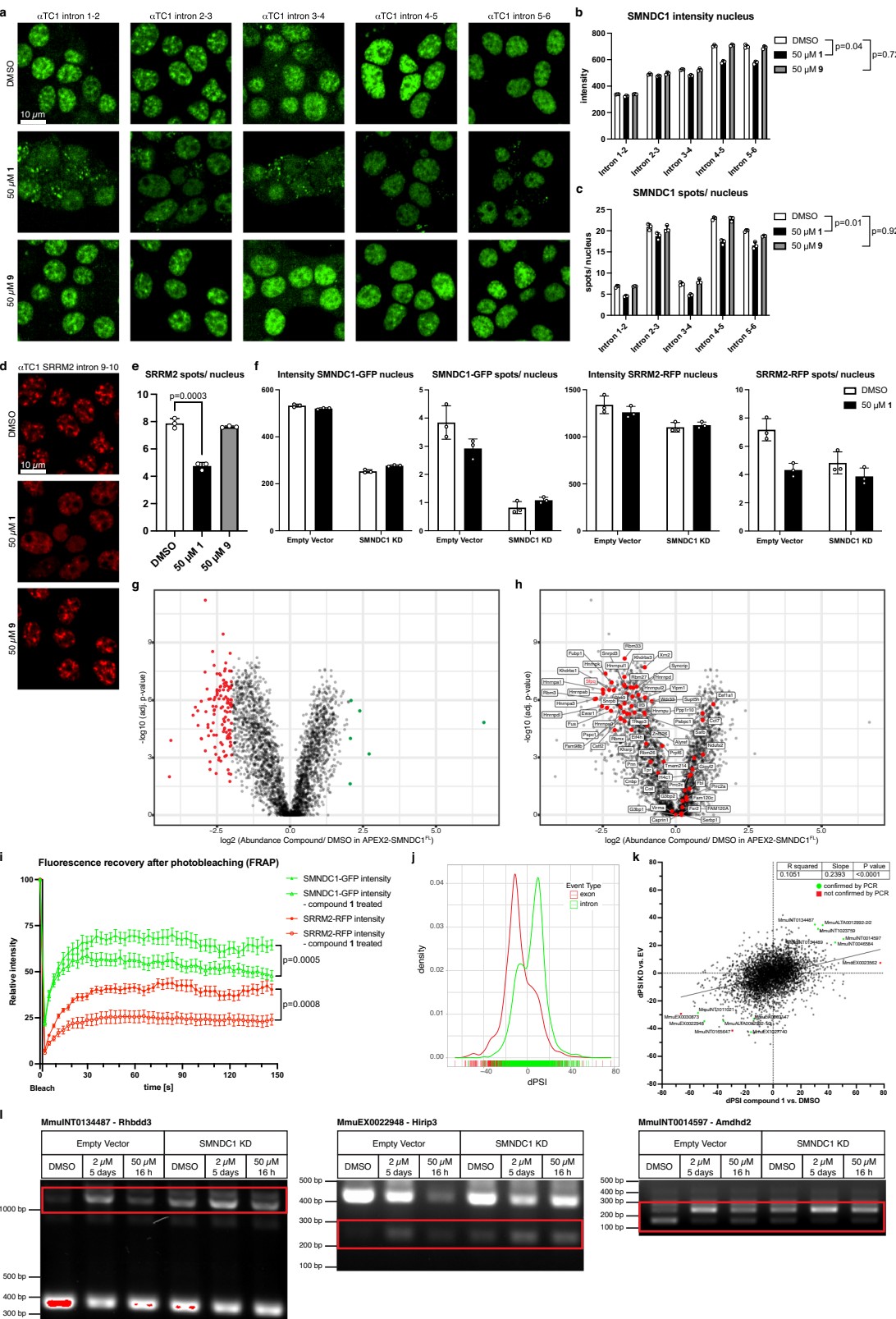

upon inhibitor treatment (Fig. 7h). The same is true for proteins with assigned localization to nuclear speckles identified in the dataset, including SRRM2 and the other main nuclear speckle organizer SON[45] (Supplementary Fig. 6b), and for proteins identified by SRSF7-APEX2[48] (Supplementary Fig. 6c). To confirm the observed effects of the inhibitor on the interactome we performed a western blot analysis of the biotin-labeled proteins after pull-down (Supplementary Fig. 6d). Also

using this orthogonal technique, we see a general loss of labeled interactors upon treatment with the inhibitor, only with APEX2-SMNDC1[FL] but not with APEX2-SMNDC1[TD]. Furthermore, we can confirm the loss of interactions to specific proteins, e.g., the sDMA-modified splicing factor SFPQ or the loss of trans-interactions to SMNDC1 itself. Interactions to SMNDC1 itself are lost both to the endogenous protein (30 kDa band with antibody against SMNDC1) and

**Fig. 7 | Cellular effects of SMNDC1 Tudor domain inhibition. a−c** Live imaging and quantifications of SMNDC1-GFP αTC1 clones treated with DMSO, compound **1** or **9**. Data shown as mean+standard deviation, $n = 3$, analyzed by ratio-paired $t$-test. **d, e** Live imaging and quantification of SRRM2-RFP spots/nucleus in cells (SRRM2-RFP intron 9-10, αTC1) treated with DMSO, compound **1** or **9**. Data shown as mean +standard deviation, two-tailed, unpaired $t$-test, $n = 3$. **f** Quantification of nuclear intensities and spots/nucleus for SMNDC1-GFP and SRRM2-RFP in live imaging data of SMNDC1-GFP (intron 2-3) SRRM2-RFP (intron 9-10) αTC1-clone treated with DMSO or compound **1** and transduced with Empty Vector (EV) or SMNDC1 knock-down (KD) plasmids. Data shown as mean ± standard deviation, $n = 3$. **g** Volcano plot showing log2 protein abundance against -log10 adjusted $p$-value (one-way ANOVA, Benjamini-Hochberg correction for multiple comparisons) of compound **1**-treated cells over DMSO control after APEX2-SMNDC1$^{FL}$ proximity labeling and biotin enrichment. 126 proteins significantly depleted (red) vs. 6 proteins significantly enriched (green), adjusted $p$-value < 0.05, |log2FC| ≥ 2. **h** Same as in **g**. Proteins with known sDMA modification marked in red and named. **i** Fluorescence

recovery after photobleaching experiment in SMNDC1-GFP intron 2-3, SRRM2-RFP intron 9-10, αTC1-clone, treated with DMSO (filled symbols) or 50 μM compound **1** (empty symbols). Relative intensity of SMNDC1-GFP (green), and SRRM2-RFP (red) in bleach region over time. Data plotted as mean with standard error of the mean, $n = 11$ for controls, $n = 13$ for compound **1** treated cells. Last timepoint analyzed by two-tailed, unpaired $t$-test. **j, k** Splicing analysis of RNA-sequencing data. **j** Alternative splicing events (exon: red, intron: green) and their density with differential percentage spliced-in (dPSI) value of compound **1** over DMSO treatment. **k** Overlapping alternative splicing events between compound **1** over DMSO treatment ($x$-axis) and SMNDC1 KD over EV ($y$-axis) with their respective dPSI-values. Results from a simple linear regression analysis, F test-derived $p$-value (F = 503.0; DFn, DFd = 1, 4285). PCR-confirmed (green) and not confirmed (red) events. **l** DNA-bands on agarose gel after RT-PCR amplification of RNA to confirm alternative splicing events. RNA isolated from αTC1 cells transfected with empty vector or SMNDC1 KD plasmid and treated with DMSO or compound **1** for 5 days (2 μM) or 16 h (50 μM).

APEX2-Fusion protein (60 kDa band with antibodies against SMNDC1 and APEX2).

As we did not observe instantaneous effects of the inhibitor on the architecture of nuclear speckles by quantifying intensity or spots per nucleus, we tested whether the inhibitor immediately influenced mobility of proteins within nuclear speckles. To this end, we applied the inhibitor **1** to live cells at a concentration of 50 μM and measured FRAP within a timeframe of 15- 45 min (Fig. 7i). Indeed, we detected a lower recovery after photobleaching for both SMNDC1 and SRRM2 when cells were treated with compound **1** while reference regions were not affected (Supplementary Fig. 6e).

To test specific cellular effects of the SMNDC1 inhibitor **1** we performed paired-end RNA sequencing and analyzed alternative splicing events using Vertebrate Alternative Splicing and Transcription Tools (VAST-TOOLS)[54]. We observed that inhibition of SMNDC1 with compound **1** led to an increased retention of introns and skipping of exons, very similar to the effects of SMNDC1 knock-down[9] (Fig. 7j). Directly comparing the differential percentage spliced-in (dPSI) values for alternative splicing events we found a significant correlation between knock-down and small-molecule inhibition of SMNDC1 (Fig. 7k). We went on to test a panel of individual events with the biggest dPSI values in both knock-down and compound **1** treatment or known to be differentially spliced upon knock-down of SMNDC1[9] (3 selected events Fig. 7l and the full panel Supplementary Fig. 6f). For the majority of events, we confirmed the expected effect of the inhibition of SMNDC1 leading to the appearance of alternative spliced isoforms, comparable to SMNDC1 knock-down. Interestingly, we could also see the effect with a long-term, low-dose treatment (2 μM over 5 days) which is more comparable to the 5 days knock-down. Combining knock-down and inhibitor treatments again did not show synergistic effects.

Overall, we demonstrated specific effects of the inhibitor on the splicing function and the localization of SMNDC1 to nuclear speckles and its proximity to interaction partners, and to the architecture of nuclear speckles in general.

## Discussion

Previous work has shown that the interaction of the Tudor domain of SMNDC1's paralog SMN with dimethylarginine causes biomolecular condensation[11]. Here, using endogenous fluorescent tags and in vitro assays, we show that also SMNDC1 undergoes phase separation. We find that SMNDC1 localizes to phase-separated membraneless organelles within the nucleus, partially overlapping with nuclear speckles. Consistent with previous findings[11] and in contrast to SMN, the SMNDC1 Tudor domain alone is not driving this condensation behavior. Rather, the protein's C-terminal IDR is sufficient for droplet formation in vitro. An RNA-binding prediction algorithm, RNAbindRplus[55], suggests that the C-terminal region, especially

residues 177−201, interacts with RNA (Supplementary Fig. 7). We therefore hypothesize that the C-terminal IDR is binding to RNAs which in turn recruit further proteins. This model is consistent with our earlier observation[9] that the majority of SMNDC1 protein interactions are lost when RNA is hydrolyzed. Likely these RNA-mediated interactions together with arginine methyl interactions mediated by the SMNDC1 Tudor domain constitute the multivalent binding platform that is a typical prerequisite in the formation of biomolecular condensates. Only the combination of the C-terminal IDR and the Tudor domain (and the nuclear localization signal) is sufficient for the correct localization of SMNDC1 and the full spectrum of interactions to both proteins and RNA. Picking apart the individual contributions of the different parts of the protein is challenging, as the C-terminal region also harbors the NLS responsible for correct organelle localization, with the regions flanking the NLS particularly disordered (Fig. 1a). From our data it is not obvious whether SMNDC1 can form nuclear droplets on its own in cellulo as a scaffold or which proteins are required for SMNDC1 to localize to pre-formed membraneless organelles as a client. However, we do observe the dissolution of nuclear speckles upon SMNDC1 inhibition or knock-down. While other factors like SON and SRRM2 are known to be important for the formation of nuclear speckles[45], these data hint at an important structural role of SMNDC1 in these membraneless organelles.

To generate chemical tools for the dose- and time-dependent study of SMNDC1 function, we focused on the protein's Tudor domain. In contrast to the C-terminal region, the Tudor domain exhibits a well-defined structure with a characteristic aromatic cage that mediates specific recognition of dimethylarginine ligands. This feature potentially enables small molecule binding often referred to as druggability. We thus set out to identify small molecule inhibitors of SMNDC1's Tudor domain using an AlphaScreen set-up. Some of the hit structures from our 90,000 compound library showed selectivity in only binding the SMNDC1 but not the SMN Tudor domain and follow up studies allowed us to derive structure-activity relationships for these compounds. Interestingly, these compounds are also active in cellular assays, although relatively high concentrations are needed. Then, the most promising inhibitor led to a loss of SMNDC1 from the nucleus and nuclear speckles and diminished SMNDC1's interaction with its partners. The kinetics we observe provide a first hint how inhibiting SMNDC1's Tudor domain might influence the architecture of nuclear speckles. While it does not immediately disrupt existing nuclear speckles, mobility and potentially inclusion of new proteins into the phase-separated compartment might be affected, leading to disruption over time. These data suggest that in cells the inhibition of the Tudor domain mediated interactions with its dimethylarginine binding partners drastically affects the protein function even with an intact C-terminal region. This perturbation then results in global splicing changes, consistent with the canonical function of the protein.

We identify specific inhibitors of SMNDC1's Tudor domain influencing SMNDC1's phase-separation behavior and splicing and architecture of nuclear speckles. These compounds are chemically distinct from inhibitors previously described for other Tudor domain proteins[56,57], namely TP53B1[58,59], Spindlin-1[60–63], UHRF1[64–66], TDRD3[67], KDM4A[68], SETDB1[69], PHF1[70], and SMN[34]. Our structure-activity relationships indicate that it is feasible to develop these compounds further to achieve specificity for SMNDC1 compared to its closest paralog SMN. Since the other ~34 human Tudor domain proteins in humans[71,72] are less conserved, we expect even lower affinities, but it will be important to conduct unbiased analyses of potential off-targets and their contribution to cellular phenotypes.

Overall, our findings will enable further studies to improve the potency and specificity of the compounds, and more deeply investigate further potential off-targets including other Tudor domain proteins beyond SMN. With more potent and specific compounds, the effect on cells and in vivo could be explored better and disentangled from unspecific toxic effects. Additionally, these compounds might be further derivatized to develop other classes of pharmacological SMNDC1 modulators and in vivo active compounds for potential therapeutic development.

## Methods

### Nomenclature
To reduce confusion due to the difference between gene and protein name we have decided to only use SMNDC1 for both.

### AlphaFold
AlphaFold[37] predictions were run via ColabFold[73] (v1.2.0) with the AlphaFold2 algorithm and the following parameters:

    msa_method=mmseqs2
    homooligomer=1
    pair_mode=unpaired
    cov=0
    qid=0
    max_msa=512:1024
    subsample_msa=True
    num_relax=0
    use_turbo=True
    use_ptm=True
    rank_by=pLDDT
    num_models=5
    num_samples=1
    num_ensemble=1
    max_recycles=3
    tol=0
    is_training=False
    use_templates=False

### Cell culture
The murine αTC1 cell line was obtained from ATCC (Cat#CRL-2934, RRID:CVCL_B036). Cells were grown in low-glucose DMEM medium (Biowest L0066) supplemented with 10% FBS, 50 U/mL penicillin and 50 μg/mL streptomycin. HAP1 cells (Horizon discovery) were grown in IMDM medium (Sigma I6529) supplemented with 10% FBS, 50 U/mL penicillin and 50 μg/mL streptomycin. The Lenti-X™ 293 T cell line was purchased from Takara Bio (632180). Cells were grown in high-glucose DMEM medium (Sigma D5796) supplemented with 10% FBS, 1 mM sodium pyruvate, 50 U/mL penicillin and 50 μg/mL streptomycin.

### Intron tagging and live imaging of cells
Cell lines with fluorescent tags in the endogenous intron loci of different genes were generated as described in Serebrenik et al., and Reicher et al.[39,74]. Cells were transiently transfected using Avalanche-Everyday Transfection Reagent with three plasmids in parallel: (1) the

donor plasmid containing the artificial intron with splice acceptor and splice donor site, the fluorescent tag GFP or RFP, and a possible correction for the frame of the targeted intron, (2) a pX330 backbone containing Cas9 and the gRNA against the donor plasmid, and (3) a plasmid expressing the gRNA against the target intron (see table below). After 3–5 days, GFP- and/or RFP-positive cells were sorted on a SONY SH800 Cell Sorter to get fluorescent single cell clones. Clones were validated for the correct integration of the intron-tag via comparison of live cell images to publicly available or in-house IF images, genomic DNA PCR amplification of the respective loci, and western blots with antibodies against the target protein and/or the fluorescent tag.

Cells were imaged on a PerkinElmer Opera Phenix automated microscope with 500 ms exposure time in either GFP or RFP channel, or on a Zeiss LSM 980 microscope. For condition-independent identification nuclear markers such as Hoechst or DRAQ5™ were used.

| Gene | Species | Intron | gRNA Sequence |
|------|---------|--------|---------------|
| SMNDC1 | mouse | 1-2 | GGACCCGTATGTTTGCCCCG |
| | mouse | 2-3 | AGACTTCCAGGCCAGCCAAG |
| | human | 2-3 | CTTGTGGAAATTGAACTATG |
| | mouse | 3-4 | TCACCTACACAGATCACGAT |
| | mouse | 4-5 | GCTAACCTGAGTTTAACCAT |
| | mouse | 5-6 | GTACCTAATGACTATTGACA |
| SRRM2 | mouse | 9-10 | GATAGCTTAATGGGCCCATG |
| SMN1 | mouse | 5-6 | TGAGCACTGGAGATACGGCG |

### Immunofluorescence
Cells were fixed in the 96-well imaging plates they were growing in before by adding 37% formaldehyde solution 1:10 to the culture medium for a final concentration of 3.7%. Cells were incubated with this for 15 min at room temperature (RT). Next, cells were washed once with PBS, followed by a 30 min permeabilization step with PBST (0.2% Tween). Afterwards, cells were blocked with a 3% BSA in PBST solution for 1 h. Primary antibodies (SMNDC1: Thermo Fisher Scientific Cat#PA5-31148; RRID:AB_2548622, 1:500; SC35: GeneTex Cat#GTX11826; RRID:AB_372954, 1:500) in 1.5% BSA in PBST were added in their individual concentrations and incubated overnight (o/n) at 4 °C. On the next day, wells were washed 3x with PBST, before incubation with secondary antibodies (Goat anti-Rabbit IgG Alexa Fluor 546 Thermo Fisher Scientific Cat#A-11010; RRID:AB_2534077, 1:500; Goat anti-Mouse IgG Alexa Fluor 488 Thermo Fisher Scientific Cat#A-11001; RRID:AB_2534069, 1:500) and DAPI (5 mg/ml, 1:2000) for 1-2 h. After 3 washing steps with PBST, cells were ready to be imaged. Cells can be stored at 4 °C before imaging on the PerkinElmer Opera Phenix automated microscope or on a Zeiss LSM 980 microscope.

### Imaging quantifications
Images were analyzed using the high-content image acquisition and analysis software Harmony® 4.9 developed by PerkinElmer. First, nuclei were identified in the channel of the nuclear marker (DAPI/ Hoechst/ DRAQ5™) (with Method C, Common Threshold 0.75, Area > 10 μm²). After the identification of nuclei, their corresponding cytoplasm was also identified using the respective nucleic acid marker (with Method A, Individual Threshold 0.15). Even though the highest staining of these nuclear markers is obviously detected in the nucleus they still produce a significant staining of the cytoplasm above background. After defining the respective cell areas, mean intensity in the different channels was measured. Finally, spots were identified with the according "Spots" algorithm (with Method A, Relative Spot Intensity > 0.053, Splitting Sensitivity: 1.0).

## Colocalization analysis

Images were preprocessed in Python version 3.7.9. Z-stacks in czi format were loaded with czifile library, version 2019.7.2, and reduced using maximum intensity Z projection. Segmentation of nuclei was carried out with Cellpose[75] (version 0.6.1) based on the DAPI/DRAQ5™ channel. Additional segmentation masks (mitotic nuclei only) were created manually. Preprocessed images and segmentation masks were saved in PNG format. CellProfiler[76] (4.0.7) was used to extract fluorescence intensity measurements for non-mitotic and mitotic nuclei separately.

All preprocessing code and the CellProfiler pipeline are available at https://github.com/reinisj/colocalization_analysis and under https://doi.org/10.5281/zenodo.8091256.

## In-vitro protein expression

Expression plasmids for SMNDC1's and SMN1's Tudor domain as used in Tripsianes et al.[7] were a kind gift from Michael Sattler. Protein expression plasmids with a GFP-fusion for droplet assays were generated by ligation independent cloning[77,78] using pET His6 GFP TEV LIC cloning vector (1GFP) which was a gift from Scott Gradia (Addgene plasmid # 29663; http://n2t.net/addgene:29663; RRID:Addgene_29663) and amplification of the respective sequences from cDNA.

BL21(DE3) competent *E. coli* cells were transformed with the respective plasmids and liquid stocks frozen at −80 °C. Volumes described here are for 450 ml total volume bacterial culture but were adjusted according to protein amounts needed. From frozen liquid stocks, 200 ml LB Kanamycin cultures were grown at 30 °C overnight, diluted with 250 ml fresh LB and grown until $OD_{600}$ reached 0.8-1. Protein expression was induced with 1 mM IPTG and bacteria grown for another 24 h at 20 °C. Bacteria were harvested by centrifugation at 4000xg for 15 min at 4 °C. Pellets were washed in 35 ml PBS and spun down again at 6000xg for 10 min at 4 °C. After removal of supernatant PBS, pellets can be stored at −80 °C.

For protein purification, pellets were resuspended in 13 ml Lysis buffer (50 mM TRIS pH 7.7, 500 mM NaCl, 1% Igepal, 2.5 mg/ml Lysozyme, 0.1 mg/ml DNase I), incubated for at least 15 min and sonicated to ensure cell lysis. Afterwards, lysates were spun down again for 20 min at 8500xg and 4 °C to remove debris pellet. In parallel, 1 ml of Ni-NTA resin (Qiagen) were added to a 15 ml tube and centrifuged at 700xg for 2 min. Supernatant was removed, and resin washed once by RIPA w/o EDTA (50 mM TRIS pH 7.7, 500 mM NaCl, 1% Igepal). Lysate supernatant was then added to equilibrated resin and rotated at 4 °C for 3 h. Beads were then spun down again at 700xg for 2 min, and washed rotating for 10 min twice by adding 14 ml RIPA w/o EDTA. Eventually, bound protein was eluted 5x with 1 ml elution buffer (250 mM Imidazole in RIPA w/o EDTA) by rotating at room temperature for 20 min, 25 min, 30 min, 45 min, o/n.

Proteins for droplet assays were then purified further by size exclusion chromatography (SEC) on a Superdex increase 200 10/300 GL column with 50 mM Tris pH 7.5, 125 mM NaCl, 10% glycerol and 1 mM DTT running buffer.

## In-vitro droplet assays

In vitro droplet assays were performed as described in Klein, Boija et al.[79]. Recombinant GFP-fusion protein purified by SEC in 50 mM Tris pH 7.5, 125 mM NaCl, 10% glycerol and 1 mM DTT running buffer was diluted to 10 µM with a concentrated PEG-8000 solution in the same buffer (and additional buffer according to protein concentration) to a final PEG-8000 concentration of 15%. In some of the experiments, total RNA isolated from αTC1 cells (10 ng/µl, Fig. 1i, k) or in vitro-transcribed RNA (100 ng/µl, Fig. 1j) was added. 10 µl of this solution were loaded onto PerkinElmer PhenoPlate™ 384-well microplates (formerly named CellCarrier Ultra microplates) and imaged immediately on the

PerkinElmer Opera Phenix automated microscope with a 63x objective at the bottom of the well.

## Fluorescence recovery after photobleaching (FRAP)

For FRAP experiments, cells harboring intron-tags in SMNDC1 and SRRM2 were seeded 24 h before imaging on a Zeiss LSM 980 microscope. 15 min before imaging, medium was changed to medium without phenol red containing DRAQ5™ 1:1000 to reduce autofluorescence and to mark nuclei. If cells were treated with compounds, these were added in the same step. After identifying a suitable cell, bleach and reference regions were defined. After taking one reference image, the bleach region was bleached 15 times for 5 milliseconds with 100% laser power at 488 nm for GFP and with 20% laser power at 546 nm for RFP. After bleaching, a new image was taken approximately every 3 s until 150 s after bleaching. Fluorescence intensities were quantified in the bleach and reference regions for every image and normalized to the intensity before bleaching.

## AlphaScreen

Compounds and controls were transferred on PerkinElmer OptiPlate-384 plates using an acoustic liquid handler (Echo, Labcyte). The AlphaScreen was conducted in 20 mM sodium phosphate pH 6.5, 50 mM NaCl, Tween 0.01%, BSA 0.1%. Protein concentration was optimized for each batch of purified protein. Optimal concentration was chosen at the lowest concentration with ~80% of maximum signal. The biotinylated binding peptide (Sequence: AAR*GR*GR*GMGR*G-NIFQKRR, $R* = sDMA$)[7] was used at 50 nM, and donor and acceptor beads at 5 µg/ml final concentration. In the first step, 10 µl of the protein solution containing either SMNDC1's or SMN's Tudor domain coupled to a 6xHis-Tag were distributed to 384-well plates pre-spotted with compounds and controls, shaken and incubated for 30 min at RT. Afterwards, 10 µl of peptide solution was added to each well and incubated for 1 h at RT. Finally, 5 µl of a solution containing both Streptavidin Donor and nickel chelate (Ni-NTA) Acceptor beads was added and again incubated for 1 h at RT. AlphaScreen signal was read out on a 2104 EnVision Multilabel Plate Reader with AlphaScreen settings, excitation time 180 ms, total measurement time 550 ms.

## Proximity labeling with APEX2

Proximity labeling with APEX2 was done following the described protocol for imaging and proteomic analysis[47]. Cell lines with a stable expression of APEX2-fusion proteins were generated using lentiviral transduction of plasmids generated with Gateway cloning of the respective fusion protein into pLEX305. pLEX_305 was a gift from David Root (Addgene plasmid # 41390; http://n2t.net/addgene:41390; RRID:Addgene_41390). The APEX2 sequence was amplified from APEX2-csGBP which was a gift from Rob Parton (Addgene plasmid # 108874; http://n2t.net/addgene:108874; RRID:Addgene_108874). Briefly, cells were incubated with 0.5 mM biotin-phenol for 30 min, after which 1 mM $H_2O_2$ was added for exactly 1 min. Afterwards the labeling reaction was quenched by 3 quick washes with Quenching solution (10 mM sodium ascorbate, 10 mM sodium azide, 5 mM Trolox in PBS). Cells were then fixed for IF analysis or detached from the plates with a cell scraper for WB or MS analysis.

## Biotin enrichment after proximity labeling

After proximity labeling, cells (~10 Mio. cells, 15 cm dish) were harvested, washed 2x in PBS, snap frozen and stored at −80 °C. Cell pellets were resuspended in 200 µL freshly prepared lysis buffer (1x PBS, 1% SDS, 2 mM MgCl₂, Protease inhibitors, Benzonase), vortexed and incubated at 37 °C for 30 min. Samples were then centrifuged for 30 min at 18,000xg and +4 °C, supernatants transferred into fresh 1.5 ml lo-bind tubes on ice. After quantification of protein amounts by Pierce™ 660 nm Protein Assay, samples were normalized to 500 µg total protein input in a final volume of 300 µl lysis buffer. For

reduction, 30 µl of 50 mM TCEP were added for a final concentration of 4.5 mM, vortexed and incubated on a shaking thermoblock at 56 °C for 1 h. After adjustment of pH by addition of 80 µl 1 M HEPES pH 7.5, 45 µl of freshly prepared 200 mM iodoacetamide were added for alkylation. Samples were vortexed and incubated on a shaking thermoblock at 25 °C for 30 min with light protection.

During reduction and alkylation 100 µl of streptavidin agarose beads (Pierce™ Streptavidin Agarose, Thermo Scientific, 20353) per sample were taken to 5 ml tubes in batches of 400 µl. To settle down the beads, tubes were centrifuged for 30 sec in a table-top spin centrifuge and settled further on ice for 3 min before taking off the supernatant. Beads were washed twice in 4 ml PBS. After the last washing, beads were resuspended in PBS and combined to a final volume of 100 µl/ sample. After distribution of 100 µl/ sample, 1.35 ml of PBS were added, and beads stored at 4 °C.

For enrichment, reduced and alkylated samples were then added to the prepared beads and rotated at 25 °C for 1 h. To settle down the beads, tubes were centrifuged for 30 sec in a table-top spin centrifuge and settled further at RT before taking off the supernatant.

BioRad Minispin columns were equilibrated on vacuum manifold with 1 ml Wash buffer 1 (0.2% SDS in PBS). Beads with enriched proteins were transferred from tubes to columns by resuspending in 2×0.5 ml Wash buffer 1. Afterwards, beads were washed 10x in 0.5 ml Wash buffer 2 (8 M Urea in PBS) and 4x in 0.5 ml PBS. After closing of columns, beads were resuspended in 2×0.5 ml digestion buffer (H₂O (HPLC grade), 50 mM Ammonium bicarbonate, 0.2 M Guanidine hydrochloride, 1 mM Calcium chloride) and transferred into fresh 1.5 ml lo-bind tubes. To settle down the beads, tubes were centrifuged for 30 sec in a table-top spin centrifuge and settled further on ice for 3 min before taking off the supernatant. 250 µl Digestion buffer were added to the beads, and beads were stored at 4 °C before the overnight digest. 10 µl trypsin (0.1 µg/µl, total 1 µg) were added to each tube at the end of the day, incubation at 37 °C rotating inside the incubator overnight (~14 h).

For solid phase extraction (SPE) stage tips were prepared as follows. 32x 1 mm in diameter C18 material was punched out from Empore C18 disk using blunt syringe needle and plunged into filter-less P200 pipette tip, pushing towards narrow end of the tip. The metal piston was pressed down to fix the C18. 24 µl oligo R3 solution (15 mg/ml in 100% acetonitrile (ACN)) were applied to the C18 tip, centrifuged at 1,000xg for 1 min inside of a collection tube. C18 was activated by washing 2x with 100 µl 100% ACN, centrifugation at 1000xg for 1 min. Columns were equilibrated with 200 µl 0.1% TFA, centrifuged at 1000xg for 30 sec, wrapped in parafilm and stored at 4 °C overnight. Right before using them for clean-up of digests the next day, C18 columns were centrifuged at 1000xg for 2 min, equilibrated again with 200 µl 0.1% TFA, and centrifuged at 1,000xg for 3 min.

After overnight digest, beads were separated via centrifugation at 1000xg for 30 sec, and complete supernatants transferred into fresh 1.5 ml lo-bind tubes. Beads were washed with 200 µl H₂O for HPLC using wide pipette tips, centrifuged again for 30 sec at 1000xg and supernatant combined with digest. The peptide samples were then acidified with 16 µl 30% TFA (~1% final) and loaded to the C18 columns in fractions of max. 250 µl, and centrifuged at 1000xg for 3 min each. After loading the full volume, columns were washed with 200 µl 0.1% TFA, and centrifuged at 1,000xg for 3 min. Samples were eluted with 2×50 µl elution buffer (90% ACN, 10% of 0.1% Trifluoroacetic acid (final 0.01%)) by centrifugation at 1000xg for 3 min. Eluates were dried in vacuum centrifuge at V-AQ, 45 °C for 1.5 h and stored at −20 °C until TMT-labeling.

Dried pellets after SPE were reconstituted in 15 µl of 100 mM HEPES pH 8.5 in H₂O for HPLC (diluted from 1 M HEPES pharmaceutical standard stock solution, pH adjusted using NaOH for HPLC). Aliquots of frozen TMT labels (Lotnr. WA314599) were equilibrated at RT for 5 min, spun down in spin-centrifuge, vortexed and spun

down again. 4 µl of respective TMTpro label were added, vortexed, spun down in spin-centrifuge and incubated at 25 °C and 300 rpm for 1 h. Reaction was stopped by adding 1.5 µl of 5% hydroxylamine solution in H₂O for HPLC (prepared fresh from 50% hydroxylamine stock solution), vortexing, spinning down in spin-centrifuge and incubation at 25 °C and 300 rpm for 15 min. Full volumes of respective TMTpro channels were then pooled into fresh 1.5 ml lo-bind tube.

For a 2D analysis, samples were fractionated by on-tip high pH fractionation. Fresh ammonium formate (AF) buffer was prepared right before using, as it is volatile: 100 mM ammonium formate in 2 ml tube (6.3 mg into 1 ml H₂O for HPLC) mixed into 4 ml H₂O for HPLC in 15 ml tube, pH 10 adjusted with two drops of 25% ammonia solution (~ 35 µl, final concentration 20 mM). For 2D analysis, 1 ml of 20 mM freshly prepared AF was added to 320 µl of pooled sample. C18 columns were prepared as described above. The eluate was loaded in fractions (max. capacity 200 µl at once), centrifuged at 1000xg for 3 min each. The column was washed with 200 µl 20 mM AF, and centrifuged at 1000xg for 3 min. Each fraction was eluted in a fresh 1.5 ml lo-bind tube. All fractionation buffers (100% ACN and 20 mM AF mixed at different ratios) were prepared fresh:

Fraction 1: Elution with 50 µl 16% ACN (24 µl ACN + 126 µl 20 mM AF), centrifuged at 1000xg for 2 min, washed with 20 µl of same buffer, collected together in tube #1, centrifuged at 1000xg for 1 min.

Fraction 2: Elution with 50 µl 20% ACN (30 µl ACN + 120 µl 20 mM AF), centrifuged at 1000xg for 2 min, washed with 20 µl of same buffer, collected together in tube #2, centrifuged at 1000xg for 1 min.

Fraction 3: Elution with 50 µl 24% ACN (36 µl ACN 114 µl 20 mM AF), centrifuged at 1000xg for 2 min, washed with 20 µl of same buffer, collected together in tube #3, centrifuged at 1000xg for 1 min.

Fraction 4: Elution with 50 µl 28% ACN (42 µl ACN + 108 µl 20 mM AF), centrifuged at 1000xg for 2 min, washed with 20 µl of same buffer, collected together in tube #4, centrifuged at 1,000xg for 1 min.

Fraction 5: Elution with 50 µl 80% ACN (120 µl ACN + 30 µl 20 mM AF), centrifuged at 1000xg for 2 min, washed with 20 µl of same buffer, collected together in tube #5, centrifuged at 1000xg for 1 min.

All 5 eluates were dried in vacuum centrifuge at 45 °C, V-AQ for at least 2 h (until dry) and frozen at −20 °C until analysis.

For a WB analysis instead of the described preparation of samples for MS, samples were not reduced and alkylated, but instead loaded on to streptavidin beads directly after lysis, quantification, and normalization. Instead of digesting proteins on the beads after enrichment, beads were transferred to lo-bind tubes with 2×0.5 mL PBS. After removal of supernatant, proteins were eluted from the beads in 3 rounds. First, 50 µl 4x LB was added, beads incubated at 95 °C for 10 mins, spun down and supernatant transferred to a new tube. Second, 50 µl 1x LB was used and combined. Last, 50 µl PBS was used and combined. Typically, 30 µl of sample were loaded on an SDS-PAGE gel.

## SDS-PAGE followed by Coomassie staining or western blotting

To separate proteins according to size, cell lysates/ protein solutions were loaded onto SDS-polyacrylamide gels (12%) with 4x Laemmli loading buffer (LB):

17.6 ml 0.5 M Tris pH 6.8
17.6 ml Glycerol
8.8 ml 20% SDS
2 ml 1% bromophenol blue
2 ml beta-mercaptoethanol

Afterwards, proteins were separated through application of an electric field (120 V for 15 min, 160 V for 90 min). For visualization of total protein, gels were stained with Coomassie Blue. To do so, the gel was fixed in fixing solution (50% methanol, 10% glacial acetic acid) for 1 h with gentle agitation. The gel was then stained in staining solution (0.1% Coomassie Brilliant Blue R-250, 50% methanol and 10% glacial acetic acid) for 20 min, followed by several rounds

of destaining with destaining solution (40% methanol, 10% glacial acetic acid).

For visualization of individual proteins, they were transferred to a nitrocellulose membrane (GE Healthcare Life Science) by electrophoresis. The membrane was blocked by 5% Milk solution in TBST for at least 1 h at RT, followed by incubation in primary antibody solution (SMNDC1: Novus Biologicals Cat#NBP1-47302; RRID:AB_10010256; SFPQ: Atlas Antibodies Cat#HPA047513; RRID:AB_2680073; APEX2 Innovagen PA-APX2-100; for all dilution 1:1000 in 5% Milk TBST) at 4 °C o/n. Membranes were then washed 3 times in TBST, followed by incubation with HRP-coupled secondary antibody solution (Peroxidase AffiniPure Donkey Anti-Mouse IgG Jackson ImmunoResearch Cat#715-035-151; RRID:AB_2340771; Peroxidase AffiniPure Donkey Anti-Rabbit IgG Jackson ImmunoResearch Cat#711-035-152; RRID:AB_10015282; Goat Anti-Chicken IgY H&L (HRP) Abcam ab97135; RRID:AB_10680105; for all dilution 1:20000 in 5% Milk TBST) for at least 1 h at RT. After 3 more washing steps, signal was detected by application of Clarity ECL Western Blotting Substrate (Bio-Rad) to the membrane with a ChemiDoc MP Imaging System (Bio-Rad) with Image Lab Touch Software Version 2.3.0.07.

## 2D-RP/RP liquid chromatography−tandem mass spectrometry analysis

Mass spectrometry analysis was performed on an Orbitrap Fusion Lumos Tribrid mass spectrometer (Thermo Fisher Scientific, San Jose, CA) coupled to a Dionex Ultimate 3000 RSLCnano system (Thermo Fisher Scientific, San Jose, CA) via a Nanospray Flex Ion Source (Thermo Fisher Scientific, San Jose, CA) interface. Peptides were loaded onto a trap column (PepMap 100 C18, 5 µm, 5 × 0.3 mm, Thermo Fisher Scientific, San Jose, CA) at a flow rate of 10 µL/min using 0.1% TFA as loading buffer. After loading, the trap column was switched in-line with an Acclaim PepMap nanoHPLC C18 analytical column (2.0 µm particle size, 75 µm IDx500mm, catalog number 164942, Thermo Fisher Scientific, San Jose, CA). Column temperature was maintained at 50 °C. Mobile-phase A consisted of 0.4% formic acid in water and mobile-phase B of 0.4% formic acid in a mix of 90% acetonitrile and 10% water. Separation was achieved by applying a four-step gradient over 151 min at the flow rate of 230 nL/min (initial gradient increase from 6% to 9% solvent B within 1 min, 9% to 30% solvent B within 146 min, 30% to 65% solvent B within 8 min and, 65% to 100% solvent B within 1 min, 100% solvent B for 6 min before equilibrating at 6% solvent B for 23 min prior to next injection). In a liquid-junction set-up, electrospray ionization was enabled by applying a voltage of 1.8 kV directly to the liquid to be sprayed, and non-coated silica emitters were used.

The mass spectrometer was operated in a data-dependent acquisition mode (DDA) and used a synchronous precursor selection (SPS) approach, which enables more accurate multiplexed quantification of peptides and proteins at the MS3 level. For both MS2 and MS3 level we collected a survey scan of 400–1600 m/z in the Orbitrap at a resolution of 120 000 (FTMS1), an AGC target was set to 'standard' and a maximum injection time (IT) of 50 ms was applied. Precursor ions were filtered according to charge state (2-6), dynamic exclusion (60 s with a ±10 ppm window), and monoisotopic precursor selection. Precursor ions for data-dependent MS$^n$ (ddMS$^n$) analysis were selected using 10 dependent scans (TopN approach). Charge state filter was used to select precursors for data-dependent scans. In ddMS$^2$ analysis, spectra were acquired using a single charge state per branch (from $z = 2$ to $z = 5$) in a dual-pressure linear ion trap (ITMS2). Quadrupole isolation window was set to 0.7 Da and collision induced dissociation (CID) fragmentation technique was used at a normalized collision energy of 35%. Normalized AGC target value was set to 200% with a maximum IT of 35 ms. During the ddMS$^3$ analyses, precursors were isolated using SPS waveform and different MS1 isolation windows (1.3 m/z for $z = 2$, 1.2 m/z for $z = 3$, 0.8 m/z for $z = 4$ and 0.7 m/z for

$z = 5$). Target MS2 fragment ions were further fragmented by high-energy collision induced dissociation (HCD) followed by Orbitrap analysis (FTMS3). The HCD normalized collision energy was set to 45% and normalized AGC target was set to 300% with a maximum IT of 100 ms. The resolution was set to 50 000 with defined scan range from 100 to 500 m/z. Xcalibur version 4.3.73.11 and Tune 3.4.3072.18 were used to operate the instrument.

## Data processing and data analysis

Following data acquisition, acquired raw data files were processed using the Proteome Discoverer v.2.4.1.15 platform, choosing a TMT16plex quantification method. In the processing step we used Sequest HT database search engine and Percolator validation software node to remove false positives with a false discovery rate (FDR) of 1% on peptide and protein level under strict conditions. Searches were performed with full tryptic digestion against the mouse SwissProt database v.2017.10.25 (SwissProt TaxID=10090, 25 097 sequences and appended known contaminants and streptavidin) with a maximum of two allowed miscleavage sites. Oxidation (+15.994 Da) of methionine and acetylation of protein N termini (+42.011 Da), as well as methionine loss (−131.040 Da) and acetylation of protein N termini with methionine loss (−89.030 Da) were set as variable modification, while carbamidomethylation ( + 57.021 Da) of cysteine residues and tandem mass tag (TMT) 16-plex labeling of peptide N termini and lysine residues (+304.207 Da) were set as fixed modifications. Data was searched with mass tolerances of ±10 ppm and ±0.6 Da on the precursor and fragment ions, respectively. Results were filtered to include peptide spectrum matches with Sequest HT cross-correlation factor (Xcorr) scores of ≥1 and high peptide confidence assigned by Percolator. MS3 signal-to-noise values (S/N) values of TMTpro reporter ions were used to calculate peptide/protein abundance values. Peptide spectrum matches with precursor isolation interference values of ≥70%, SPS mass matches ≤65% and average TMTpro reporter ion S/N ≤10 were excluded from quantitation. Both unique and razor peptides were used for TMT quantitation. Isotopic impurity correction was applied. Data were normalized on total peptide amount for correction of experimental bias and scaled 'on all average'. Protein ratios are directly calculated from the grouped protein abundances using a one-way ANOVA hypothesis test, followed by Benjamini-Hochberg correction for multiple comparisons. The mass spectrometry proteomics data have been deposited to the ProteomeXchange Consortium via the PRIDE partner repository with the dataset identifier PXD037092 and 10.6019/PXD037092.

## Enrichr analysis

Gene symbols of proteins identified by mass-spectrometry or subsets thereof were analyzed for enrichment of gene ontology (GO) Biological Process 2021 terms with the online tool "Enrichr" (https://maayanlab.cloud/Enrichr/)[80]. For each term that is associated with at least one identified gene symbol an odds ratio was calculated based on the size of the input dataset and the size of the group of gene symbols associated with that term. Furthermore, a p-value, and an adjusted p-value (Benjamini-Hochberg method for correction for multiple hypotheses testing) was determined.

## NMR experiments

Isotope-enriched SMN$_{84-147}$ and SMNDC1$_{65-128}$ were expressed and purified as described in Tripsianes et al.[7]. NMR experiments were performed on Bruker Avance III spectrometers operating at 600 MHz or 800 MHz $^1$H frequencies using H/N/C triple-resonance cryogenic probes. All NMR acquisition was performed in 3 mm tubes at 25 °C. Spectra were processed using Topspin 3.5 (Bruker) and analyzed with Cara 1.9.17[81] or NMRglue[82]-based Python scripts. Chemical shift assignments were transferred from Selenko et al.[83] and Tripsianes et al.[7] (BMRB: 4899 for SMN, 18006 for SMNDC1). All titration

measurements were performed in 20 mM sodium phosphate buffer pH 6.5, 50 mM NaCl, 4 mM dithiothreitol and 10% (v/v) $D_2O$ as deuterium lock. Aqueous, buffered inhibitor stock solutions of maximal 20 mM concentration were prepared and carefully adjusted to pH 6.5. Inhibitor concentration was measured by addition of 100 μM DSS, peak integration and calculating with $C = \frac{I \cdot N_{DSS} \cdot C_{DSS}}{I_{DSS} \cdot N}$ where C, I, N and $C_{DSS}$, $I_{DSS}$, $N_{DSS}$ is the concentration, peak intensity, and number of protons of inhibitor and DSS, respectively. Titration experiments were performed with 50 μM $^{15}$N-labeled $SMN_{84-147}$ and $SMNDC1_{65-128}$ and addition of 0, 0.05, 0.1, 0.25, 0.5 and 0.8 mM compound **13**. Since the titration did not reach saturation, an additional point was measured for SMNDC1 with 8 mM compound **13**. An apparent dissociation constants $K_D$ of around 1 mM was calculated from CSP data using $CSP = \frac{CSP_{max} \cdot c}{c + K_D}$ with c being the concentration of compound **13** and $CSP_{max}$ the CSP at saturation. However, this value is obstructed by solubility issues of **13** and therefore not comparable with the IC50 values from the AlphaScreen assay, where lower concentrations were used. To record intermolecular NOE data, a 1 mM sample of $^{13}$C,$^{15}$N-labeled $SMNDC1_{65-128}$ was prepared, lyophilized, and resolved in $D_2O$ containing 20 mM buffered compound **13**. To confirm saturation of binding, a $^1$H,$^{15}$N-HSQC spectrum was acquired and compared with the titration data. $\omega_1$-$^{13}$C-filtered and $\omega_2$-$^{13}$C-filtered two-dimensional NOESY and $\omega_2$-$^{13}$C-filtered, $\omega_1$-$^{13}$C-edited three-dimensional NOESY experiments[84] were recorded with 150 ms mixing times. Chemical shift assignments were transferred from Tripsianes et al.[7].

## Docking calculation

Docking calculations were performed using the HADDOCK webserver[52,53]. Structure and topology files for compound **13** were generated by prodrg2[85]. The SMNDC1/sDMA structure (PDB: 4A4H)[7] was used as a protein model with the sDMA removed beforehand. Instead of defining active/passive residues, intermolecular NOE contacts were introduced as ambiguous restraints. Visible and assigned NOE crosspeaks were defined as distance restraints with a lower limit of 0.5 Å and upper limit of 5 Å. Peak intensities of crosspeaks were measured and normalized to the strongest peak. According to their relative intensities, upper distance limits were gradually lowered to 3.5 Å for non-overlapping crosspeaks. 1000, 400 and 400 structures were calculated for the different stages of rigid body docking, semi-flexible refinement, and final refinement, respectively. Parameters were chosen as suggested by HADDOCK for small molecule docking. 200 structures were analyzed and clustered by RMSD with a cutoff of 1.5 Å resulting in one cluster, which contains all analyzed structures.

## SMNDC1 knock-down

SMNDC1 knock-down was performed as described in Casteels et al.[9]. Briefly, Smndc1 shRNA from the TRC shRNA library (https://portals.broadinstitute.org/gpp/public/) (TRCN0000123795) was cloned into pLKO.1 (Addgene plasmid #10878). This plasmid was packaged into lentivirus in Lenti-X™ 293 T cells (BOSC-23, RRID:CVCL_4401, TakaraBio Cat#632180) with Lipofectamine™ 3000 (Thermo Fisher Scientific L3000008) and packaging plasmids psPAX2 (Addgene plasmid #12260) and pMD2.G (Addgene plasmid #12259). Target cells were transduced with viral supernatant after filtering and addition of 8 μg/ml Polybrene® (Santa Cruz Biotechnology sc-134220) 48 h after transfection. Medium was changed 24 h later.

## Splicing PCRs

To perform splicing PCRs, RNA was isolated from pelleted cells using the RNeasy Mini Kit (Qiagen, #74106). RNA was then reverse transcribed with LunaScript RT SuperMix Kit (NEB #E3010). cDNA was PCR-amplified with OneTaq® Quick-Load® 2X Master Mix (NEB #M0486) for 35 cycles with the following primers as suggested on vastdb.crg.eu:

| Gene | Event | Orientation | Sequence |
|---|---|---|---|
| Rhbdd3 | MmuINT0134487 | F | TTCCTGCACAACTCCACTGTG |
| Rhbdd3 | MmuINT0134487 | R | GCCAGAGACTTGCAAAGGACA |
| Hirip3 | MmuEX0022948 | F | AGGCAGCAGTAATGGTGACAG |
| Hirip3 | MmuEX0022948 | R | GCGACACTTCTCCAAGGAAGG |
| Amdhd2 | MmuINT0014597 | F | GCCTGGCTTTATCGATGTGCA |
| Amdhd2 | MmuINT0014597 | R | TGTGATAAACCTCTGGTGGGGA |
| Vps37a | MmuEX1027740 | F | AGGCAAAAGGCAAACCGTTTT |
| Vps37a | MmuEX1027740 | R | TGTTCTCTTTTCCTTGAAGCTATTGA |
| Taf15 | MmuEX0046179 | F | ATGACCGTCGTGATGTGAGTA |
| Taf15 | MmuEX0046179 | R | CAGCATCTGGTCTGGGTCCAT |
| Dgat1 | MmuINT0048631 | F | TGGGTTCCGTGTTTGCTCTG |
| Dgat1 | MmuINT0048631 | R | CGGTAGGTCAGGTTGTCTGGA |
| Gnb1 | MmuEX0021329 | F | GACCAGCCTCGCCGACTC |
| Gnb1 | MmuEX0021329 | R | GCAGCTGGTCAAGTTCACTCA |
| Strada | MmuEX0045528 | F | TCTTGTAAGTAAACCAGAGCGCA |
| Strada | MmuEX0045528 | R | GGTGAGCAGCTCATAACACCC |
| Abcc8 | MmuEX0003147 | F | TACGAGGCCCGGTTCCAG |
| Abcc8 | MmuEX0003147 | R | AGCCCCTCATAGCTCTCTGC |
| Syt7 | MmuEX0046044 | F | GGCAAACGCTACAAGAATTCCT |
| Syt7 | MmuEX0046044 | R | GCATCTCGCTGGTAAGGGA |
| Agfg1 | MmuEX0004229 | F | CTGCTCAGACACAACCTGCTT |
| Agfg1 | MmuEX0004229 | R | CTGTCTGCTGAGGGAAAGCTG |
| Napb | MmuEX0030873 | F | AAACTCCACATGCAGCTCCAG |
| Napb | MmuEX0030873 | R | TCGGCAATGGTGATATGGTGC |
| Pdss1 | MmuALTA0012992-2/2 | F | GAGCTGCACATCTCCACCAGA |
| Pdss1 | MmuALTA0012992-2/2 | R | TGGATCATTTCTGCAACTAAGGCT |
| Gm10451 | MmuINT1011021 | F | CACTGTGGCCAAACATCCCTG |
| Gm10451 | MmuINT1011021 | R | TGGATCAAGATGTTGCAATTTTTATC |
| Trmt2a | MmuINT0165647 | F | AGGTGAAGAGAGTAGTGGGAA |
| Trmt2a | MmuINT0165647 | R | CGTGGTGGGTCTAGAACAGCT |
| Dalrd3 | MmuINT0046584 | F | ATCCTCTCCGTGGCTACCATC |
| Dalrd3 | MmuINT0046584 | R | TACAGACCTTGCTCCGTTCCA |
| Icmt | MmuEX0023562 | F | TCAGAGCTTGTTTCCTTGGCT |
| Icmt | MmuEX0023562 | R | GGCCAGAAGATGTTCTCGAGC |
| Mus81 | MmuINT1023759 | F | AGCCTTCCACAAACCCTCTCT |
| Mus81 | MmuINT1023759 | R | GGTGCTGTATCGATCCACCAC |

The PCR products were run on a 1% Agarose gel for 30 min at 100 Volt.

## RNA sequencing and transcriptome analysis

RNA sequencing libraries were prepared from low-input samples using the Smart-seq2 protocol[86]. The subsequent library preparation from the amplified cDNA was performed using the Nextera XT DNA library prep kit (Illumina, San Diego, CA, USA). Library concentrations were quantified with the Qubit 2.0 Fluorometric Quantitation system (Life Technologies, Carlsbad, CA, USA) and the size distribution was assessed using the Experion Automated Electrophoresis System (Bio-Rad, Hercules, CA, USA). For sequencing, samples were diluted and pooled into NGS libraries in equimolar amounts.

Expression profiling libraries were sequenced on NovaSeq 6000 instrument (Illumina, San Diego, CA, USA) with a 100-base-pair, paired-end setup. Raw data acquisition and base calling was performed on-instrument. Subsequent raw data processing off the instruments involved two custom programs (https://github.com/epigen/picard/) based on Picard tools (2.19.2) (https://broadinstitute.github.io/picard/). In a first step, base calls were converted into lane-specific, multiplexed, unaligned BAM files suitable for long-term archival (IlluminaBasecallsToMultiplexSam, 2.19.2-CeMM). In a second step, archive BAM files were demultiplexed into sample-specific, unaligned BAM files (IlluminaSamDemux, 2.19.2-CeMM).

NGS reads were mapped to the Genome Reference Consortium GRCm38 assembly via "Spliced Transcripts Alignment to a Reference"

(STAR)[87] utilizing the "basic" Ensembl transcript annotation from version e100 (April 2020) as reference transcriptome. The mm10 assembly of the UCSC Genome Browser was used for downstream data processing, and the Ensembl transcript annotations were adjusted to UCSC Genome Browser sequence region names. STAR was run with options recommended by the ENCODE project. NGS read alignments overlapping Ensembl transcript features were counted with the Bioconductor (3.11) GenomicAlignments (1.24.0) package via the summarizeOverlaps function in Union mode, ignoring secondary alignments and alignments not passing vendor quality filtering. Since the Smart-seq2 protocol is not strand specific, all alignments irrespective of the gene or transcript orientation were counted. Transcript-level counts were aggregated to gene-level counts and the Bioconductor DESeq2[88] (1.28.1) package was used to test for differential expression based on a model using the negative binomial distribution.

### Splicing analysis

Alternative splicing events were characterized and quantified using VAST-TOOLS[54] (2.5.1) in conjunction with the *Mus musculus* database (vastdb.mm2.23.06.20), based on the Genome Reference Consortium assembly GRCm38.p5 and Ensembl transcript annotation 88 (March 2017). Briefly, NGS reads were aligned for each read group independently, read groups were merged into samples and samples were combined into a summary table. The differential splicing events were called via the VAST-TOOLS "compare" algorithm (min_dPSI > 15, min_range > 5) and further filtered for genes showing statistical significance (adjusted *P*-value ≤ 0.1) and a sizable effect (absolute log2-fold change ≥ 1.0) in the differential expression analysis.

### Statistics and Reproducibility

For representative images, these are the number of replicates with similar results:

Figure 1b: $n = 5$, Fig. 1c: $n = 4$ (see Supplementary Fig. 1c), Fig. 1d: $n = 5$, Fig. 1e: $n = 5$, Fig. 1f: $n = 3$, Fig. 1g: $n = 3$, Fig. 2a: $n = 3$, Fig. 2b: $n = 3$, Fig. 2d: $n = 3$, Fig. 2e: $n = 3$, Fig. 2f: $n = 3$, Fig. 3b: $n = 2$, Fig. 7l: $n = 1$, Supplementary Fig. 1d: $n = 1$, Supplementary Fig. 2b: $n = 2$, Supplementary Fig. 3a: $n = 2$, Supplementary Fig. 5g: $n = 3$, Supplementary Fig. 6f: $n = 1$.

### Reporting summary

Further information on research design is available in the Nature Portfolio Reporting Summary linked to this article.

## Data availability

The mass spectrometry proteomics data have been deposited to the ProteomeXchange Consortium via the PRIDE partner repository with the dataset identifier PXD037092 and 10.6019/PXD037092. NMR structures have been deposited to PDB with the identifier 8POI and to BMRB with the identifier 34831. RNA-seq data have been deposited to GEO with the identifier GSE231600. Source data are provided with this paper.

## Code availability

All preprocessing code and the CellProfiler pipeline for the colocalization analysis are available at [https://github.com/reinisj/colocalization_analysis] and under the https://doi.org/10.5281/zenodo.8091256 [https://doi.org/10.5281/zenodo.8091256][89].

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

## Acknowledgements
We thank CeMM's Molecular Discovery Platform for help with small molecule screening and generation and processing of proteomics data and CeMM's Biomedical Sequencing Facility for generation and processing of our RNA-sequencing data. Yuchen Wu assisted with clone-picking and computational docking. We acknowledge NMR measurements at the Bavarian NMR Center, TUM, Garching. Lennart Enders gratefully acknowledges support by a DOC PhD Fellowship of the Austrian Academy of Sciences (25721). This work was supported by the Deutsche Forschungsgemeinschaft (DFG, German Research Foundation)—SFB 1309—325871075 (M.S). Research in the Kubicek lab is supported by the Austrian Federal Ministry for Digital and Economic Affairs and the National Foundation for Research, Technology, and Development; the Austrian Science Fund (FWF) F4701; and the European Research Council (ERC) under the European Union's Horizon 2020 research and innovation program (ERC-CoG-772437).

## Author contributions
L.E. and S.K. planned the study and designed the experiments; L.E. performed most of the experiments with help from A.K., M.M., T.T., A.Re. and T.C.; M.Si. performed chemical syntheses and SAR studies; J.B. and S.G. performed NMR experiments; E.H. and A.Ru. performed biotin-purification and MS sample preparation; L.E., J.R., M.S., A.K., M.M. and A.Ru. analyzed the data; L.E. and S.K. wrote the manuscript with input from all co-authors; C.B., G.E.W., J.T.H., M.Sa. and S.K. supervised the work.

## Competing interests
L.E., M.Si. and S.K. are filing a patent based on the findings in this manuscript. The other authors declare no competing interests.
