## [Peer Review File · Nature Communications]

REVIEWER COMMENTS

Reviewer #1 (Remarks to the Author):

The manuscript by Enders et al. aims to characterize phase separation properties of SMNDC1 and identify small molecules using an AlphaScreen to impair this phase-separation property. Furthermore, the authors present the structure of the SMNDC1-Tudor domain in complex of one of the identified inhibitors. In general, the presented study is of high interest and might be suitable for Nature Communications. Nevertheless, despite the impressive amount of work the manuscript at the current state fails to be entirely convincing as outlined by the different points listed below. In addition, the manuscript reads at the moment at instances still as a sum of different parts and not one coherent text, which the authors might also want to address.

Major Points:

- The readability of the introduction should be improved as currently the paragraph about the Tudor domains (lines 70–75) seems to be oddly placed and not well integrated with the rest of the text.
- As the authors analyzed the intrinsic disorder as well as used the AlphaFold model it looks like a miss that they did not connect these in a better way (Fig. 1A; lines 80-93). For example does the disorder predictor correctly indicated the helices found in the AlphaFold model and based on how many co-evolution constraints did the AlphaFold model base the helix:helix interaction on. A better labeling of the AlphaFold model is also needed as at the moment it is not entirely clear where the N-terminal and C-terminal extensions are. It might be beyond the scope of the current manuscript it would be highly informative to have some experimental data on the full-length protein such as CD spectra, to confirm the existence of the helices in the rather unstructured regions, as AlphaFold tends for some IDPs (e.g. Tau40) to indicate helical content, which experimentally could not be verified.
- o The additional benefit of the different AlphaFold structures with the GFP inserted in the intron regions is not clear to this reviewer and they seem to overcomplicate the current figure one.
- Figure 1 in vitro droplet formation: It was a bit unclear how the authors actually performed these assays, whereas the figure legend and the main text (lines 112–124) indicated the use of either PEG-8000 or RNA in the methods section (lines 635–643) the author only mention the usage of PEG-8000. It should be clarified how the experimental set-up was.
- The authors should reconsider the set-up of figure 4 as it is hard to assess at the moment. Maybe putting compound 1 in the center enlarged and rearranging the different modifications around would make this figure more accessible.

- The structures of compound 13 bound to SMNDC1 in Figure 5a and 5b look slightly different, which is maybe due to showing the protons of the H₂N group in panel A. Furthermore, as panel B seems to be slightly rotated this should be indicated.
- Figure 5C, in the NMR spectrum are about four additional peaks experiencing large chemical shift changes that were not annotated. A detailed reasoning for the resonances chosen to be labeled would likely resolve this.
- Figure 5D, it is definitely an important panel for the detection of the intermolecular NOEs. In slight contrast to the shown panel, which indicates the unspecific assignment of the ring-protons of the aromatic ring protons, the list of the intermolecular NOEs indicates stereospecific assignment for the ring-protons.

Minor Points

- Line 134 – abbreviations should be introduced, thus it is missing for IF.
- Lines 149-151 – the sentence ending with ...which are often the same proteins. Remains unclear. Same proteins identified in different datasets or do the authors mean the same proteins bind symmetrical and asymmetrically methylations. Please clarify.
- Line 398 – *In vitro* should be in italic
- Figure 2c legend: The details of the Enrichr analysis here and in the maintext (line 146) should rather be presented in the methods section.
- Line 420 – adj. should read adjusted
- Lines 427-430 – The text describing panels c and d seems to have a couple of grammar issues
- Line 430 – should readhits marked in yellow....
- Suppl. Figure 4 – the orientation of Panel A in relation to Fig. 5 should be indicated. In Panel C some grey ticks are shown, which seems to be an artifact.
- Suppl. Figure 5k – The RNAbindRplus Score could possibly related to the AlphaFold structural model
- Suppl. Table 2 – In some of the compound cells red triangles are shown, which are not discussed and are possibly an artifact
- Suppl. Table 4 – some values in the HADDOCK table need clarification and maybe discussion in the main text: why is the error of the HADDOCK score 0? Same for the z-score. Also the RMSDs of 0.6 ± 0.6 (overall) and especially 0.61 ± 0.788 (pairwise) with the error being larger than actual value look suspicious.
- Line 613 – *E.coli* should be in Italic

- Lines 618-623: 4000g should be 4000xg to be consistent with other parts of the methods (see e.g. lines 713–727)
- Lines 728/733/739 –H₂O should be H₂O
- Lines 773–794 – As SDS PAGE is a standard method and the others did not seem to use any particular special approaches the composition of the Lämmli buffer might not be necessary to provide. Rather important details like the used voltage, run length and type of the used Gels are on the other hand not provided.
- Line 876 – should likely read....saturation, an additional point...
- In general the references need some clean-up as several different styles were used...

Reviewer #2 (Remarks to the Author):

This is an interesting paper linking Tudor domain interactions with dimethyllysine, and intrinsically disordered protein (IDP) interactions with RNA binding as two key molecular interactions that drive liquid-liquid phase separation (biomolecular condensates; membrane-less organelles). The study reports new chemical compounds that bind to the aromatic cage of SMNDC1 Tudor domain and appear to inhibit the interaction with sDMA, with consequent mildly reduced number of GFP-positive signal in phase separated nuclear speckles.

However, this compound is of low affinity and therefore difficult to use for drawing firm conclusions. Overall the manuscript would benefit from the use of additional controls (such as suggested below) and better quantification of selected binding affinities by a complementary method other than just AlphaScreen assay (ideally with recombinant protein), and more consistent use of construct/protein nomenclature. Specific suggestions for improvements are listed below.

1. Fig 1. Placement of GFPs within 3D structure is hard to see within the alpha fold simulated structures. Instead annotation of where the GFP insertions are on the WT alphafold prediction would be best, and then use a schematic to depict the fusion proteins instead of complicated alpha fold structures. Authors can include the current AF structures in supplemental materials for readers who wish to see.
2. Intron2-3 GFP insertion appears to disrupt the small (predicted) alpha helix, but this is the construct used throughout most of the paper. Have the authors compared several of the phenotypes of Intron2-3 with the other insertions to show that 2-3 is really representative and not a fluke?
3. Nomenclature: ‘SMNDC-GFP Intron2-3’ is exemplified in Fig 1C, but then all remaining nomenclature is ‘SMNDC1-GFP’. Are all subsequent expts done with the Intron 2-3 insertion? This ambiguity should be fixed.
4. Nomenclature: Is ‘truncated SMNDC1-APEX2’ (in text) the same as SMNDC1-TudorDom-APEX (Fig2b)? Nomenclature should be consistent or each term better defined.

5. Nomenclature: Does the truncated protein correspond to one of the constructs indicated in Fig1a? It would be good to map this construct (and others named, including that used for NMR) onto Fig 1A and include the protein names in Fig 1A.
6. The observation that SMNDC1 itself was enriched in full length over truncated SMNDC1-APEX2 could be due to fewer lysine-containing sequences available to be biotinylated in Tudor alone.
7. Pg 8. To conclude (suspect) that “the Tudor domain is responsible for a major part of SMNDC1’s specific interactions”, the better experiment would have been to use a construct with only the Tudor domain deleted. The authors had concluded that the C-term (post-Tudor sequences) drives droplet/speckle formation/localization, and the Tudor alone was less restricted to speckles. It does not follow necessarily that the Tudor harbors all the specificity. Only a small subset of the full length APEX2 interactome were SDMA proteins. This logic does not follow.
8. Pg 8: “with IC50 values of 3 to 20 μM ” should be qualified as “in the range of ...” or ‘low micromolar’, as IC50 values from a 4pt titration are not at all accurate.
9. 50uM is a high concentration for cellular studies. At this concentration it is likely that other unintended proteins may also be substantially affected. Indeed, the authors state that at longer time points cells start to die. Can the authors be certain that the molecular events they report, such as loss of SMNDC1 from the nucleus is due solely to inhibition of Tudor domain-sDMA binding? As a control for off target effects, the authors could use one of the inactive, but chemically similar compounds (including similar predicted cell permeability properties), from the SAR series. The inactive compound should not show any changes in protein localization at 50uM.
10. In the discussion, the authors state: Likely these RNA-mediated interactors together with arginine methyl interactions mediated by the SMNDC1 Tudor domain constitute the multivalent binding platform that is a typical prerequisite in the formation of biomolecular condensates. Given the confounding issues of using very high compound concentrations, further support for this notion could be provided by mutation of key aromatic cage residues in the Tudor domain. If the mutations phenocopy inhibitor treatment, this would also help validate compound 1 as a functional Tudor domain inhibitor in cells.

Reviewer #3 (Remarks to the Author):

The manuscript “Pharmacological perturbation of the phase-separating protein SMNDC1” establishes that SMNDC1 partially co-localizes into nuclear speckles, which has been also proved by analysis of its ligands. SMNDC1 is the name of the gene encoding SPF30 protein (<https://www.uniprot.org/uniprotkb/O75940/entry>), a splicing factor with roles in spliceosome assembly and function. We suggest that SPF30 be used throughout the manuscript to refer to this protein to avoid confusion with SMN (Survival Motor Neuron protein). We are concerned that the low quality of the images that should define the compartment to which SPF30 localizes prevents a clear understanding if these are standard nuclear speckles, a subset, or a different but overlapping compartment. To answer these questions, immunofluorescent analysis of fixed cells by confocal, high resolution wide-field fluorescent light microscopy, or even STED should be used – the only microscopy

documented in the manuscript is live imaging, so it is not clear how this analysis has been done. There are also several instances where methodology is not sufficiently described to allow reproduction. The co-localization of SPF30 protein into nuclear speckles and its overall function in splicing may be specifically disrupted by a new chemical inhibitor. Although tudor domains are increasingly recognized for their role in biomolecular condensation, chemical inhibitors of specific tudor domains/tudor domain containing proteins have so far been lacking. This brings impact to the present manuscript. However, inhibition of SPF30's role in splicing – its best characterized function – has not been adequately tested. Thus, the characterization of the effects of disrupting SMNDC1 function must be improved. Below, we make constructive suggestions for improving the manuscript and point out a number of specific weaknesses that prevent publication of the manuscript in its current form.

Detailed comments:

1/ Figure 1 is hard to understand because it is too big and complicated and there are multiple conclusions. We think this should be split in half (new fig 1 and 2), perhaps separating the in vivo from the in vitro experiments. This would enable the authors to describe more rigorously the GFP-truncated proteins in the text as well as in the figure legend.

Line 95 – The authors claim that all structural elements are predicted normally. However, the protein with disrupted Tudor domain is obviously expressed less than other protein versions (Suppl. Fig. 1a). Please reflect this evidence in the main text, ideally by quantifying the tagged versus endogenous protein in 3 replicates. Regarding the Suppl. Fig. 1b, could the authors show a different field of cells overexpressing CLK1? It is difficult to interpret the staining relative to cell morphologies.

Fig. 1d – The colocalization of SMNDC1 with Mitotic Interchromatin Granules must be quantified, as well as the “partial” co-localization of SC-35 and SMNDC1 in interphase. Was this partial co-localization also observed in the case of Mitotic Interchromatin Granules, or did they co-localize completely during mitosis?

“Mitotic Interchromatin Granules” is only used once, so we suggest getting rid of the acronym MIG.

Fig 1f – Hexanediol is really a dubious way of implicating LLPS as the underlying process, due to other effects of hexanediol on cells. For example, it inhibits kinases and phosphatases (<https://pubmed.ncbi.nlm.nih.gov/33814344/>)! A much better way of addressing reversibility in this system would be FRAP, which would also be more easily quantifiable with more interpretable parameters like % recovery rather than intensity per nucleus or spots/nucleus, which are more ambiguous.

Fig. 1g – The overexposed GFP images are not explained. It is preferable to have the same microscope setting in all conditions.

Fig. 1k – Please add information about the presence of RNA into the figure for better understanding.

2/ Figure 2:

Fig. 2b – The image quality is very low. It appears that the APEX2 construct fills the entire nucleus, as does the streptavidin signal. This could account for the large number of proteins identified. Could you show the cell boundary to provide us with better information about the expressed proteins in the cytoplasm?

Fig. 2d – Could you please clarify your statement on sDMA proteins on lines 147-149, since there is only an overlap of 67 proteins?

Fig. 2h – SRSF7-APEX2 served as a control nuclear speckle protein. We think this should be removed, because there is no negative control for the SRSF7-APEX experiment. Also, the information provided is not directly related to the main argument.

In the text related to Figure 2, the specificity of SMNDC1 towards sDMA and aDMA, which was published in Tripsianes, Nature Structural & Molecular Biology, 2011, should be mentioned because it is important to the interpretation.

The main issue of the manuscript is that information on SMNDC1 ligands (Figure 2) is not fluently connected to perturbation of SMNDC1 function. Thus, the motivation for perturbing SMNDC1 function is disconnected from the results so far.

3/ Figure 3:

Fig. 3f-l – Please include the IC50 for SMNDC1 and SMN1 in the figures for better orientation.

Please explain the abbreviation SAR on line 204.

4/ Would it be possible to display a visualization of your statement of “the β 2-strand” from lines 259-260 in Figure 5? The position of this strand is not reflected in the figure. The authors should also elaborate on the significant chemical shifts in Figure 5 for a broader audience that could be unfamiliar with NMR.

5/ Please refer to the compounds with a name like “compound 1” or even C1 for short. Please do not refer to the compound in the sentence as “1”. This is far too confusing.

6/ Figure 6:

This figure is not well laid out. It could be split into two figures or certain panels could be moved to the supplement. The fonts are too small to see even allowing the figure to take up the whole page.

The canonical function for SPF30 is in splicing. The data asking if the inhibitor shows predicated changes in splicing are completely unacceptable. We think this experiment should be done globally and the chemical inhibition should be compared to and SPF30 knockdown or depletion by degron. High quality, low cycle number, quantified RT-PCR could be an option in the absence of global data. Another option

would be to add the compounds to an in vitro splicing reaction and show dose-dependent effects on splicing. Instead, we are shown completely saturated images of bands for which there are no proper positive or negative controls and the purported results are not believable (panel g). It is NOT convincing that splicing is inhibited.

The validity of the statistical analysis of the graphed data in this figure (panels b, c, e, and f) is questionable. Various parameters are analyzed by ratio paired t-test from live cell imaging. What are the numbers of cells and numbers of replicates considered?

Fig. 6a – Please show DAPI staining to indicate the limits of the nuclei and to show if cells are intact and/or in mitosis.

Fig. 6h&i – The volcano plot is very confusing. We do not understand the expectations regarding the volcano plot. If the compound is inhibiting binding to DMA ligands, why not display an analysis of the DMA ligands indicated in Fig 2? Why not do a co-IP from treated versus control cells and ask if the amount of DMA-modified proteins that co-IP change? An analysis of annotated hits, such as STRING analysis, should be considered.

6/ Supplementary Figure 3:

Fig. 3b-c The authors should explain how the titration unit of protein-peptide, which is in 'nM', relate to the titration unit of beads, which is in 'µM/ml'. The relationship between these two titration experiments is not clear.

7/ Conclusions regarding cell death based on Supplementary Figure 5:

Lines 291-293 say that “the majority of cells was not apoptotic or necrotic (necrotic refers to a tissue – this word should be removed and possibly be replaced with the descriptor that is more relevant to the authors) as shown by Annexin V or Propidium Iodide staining (Suppl 5a).” We disagree with the interpretation. Based on the error bars, the differences shown in Suppl. Fig. 5a are significant. The plots seem to indicate that compound 1 leads to 2-3 times more AnnexinV or PI staining, respectively. How a snapshot in the staining experiment relates to the overall number of cells that are dying is not obvious. If the authors wish to monitor the number of living cells in the culture, perhaps they should use mitotracker or MTT to identify and analyze remaining living cells in image data.

8/ Discussion

The pharmacological relevance of your study, the significance of obtaining SMNDC1 interactome, the role of SMNDC1 in condensation should be more rigorously discussed. More recent publications should be included and include more recent publications.

Reviewer #4 (Remarks to the Author):

SMNDC1 (SPF30) is a component of the human spliceosome that interacts with dimethyl-arginines of Sm proteins of spliceosomal snRNPs. The authors describe the identification of compounds that bind to the dimethyl-arginine-binding pocket of SMNDC1 and, in some cases, distinguish between the pockets of SMNDC1 and the closely related protein SMN. Such compounds could be important for the future investigation of the importance of SMNDC1 for gene regulation and the determination of cellular identity. The authors also provide evidence that SMNDC1 phase separates and occupies nuclear bodies that partially overlap with nuclear speckles known to contain splicing components, as well as evidence that their compounds might disrupt SMNDC1 function. While I agree with the authors that the identification of potential small molecule inhibitors of SMNDC1 might well be important, I think that the data they provide about the normal behavior of SMNDC1, as well as the ability of their compounds to disrupt its function, has numerous flaws, as described below, that should preclude the publication of their manuscript in its present form.

1. In Fig. 2b the authors look at full length SMNDC1, as well as SMNDC1 containing only its Tudor domain, both fused to APEX, and examine SMNDC1 localization, as well as biotinylation by binding to streptavidin. As the authors point out, part of the Tudor-only construct is in the cytosol, presumably because it lacks its NLS. However, the large portion in the nucleus appears to be in the same phase-separated regions as the full length protein. This is a contradiction with the literature, where Tudor-only SMNDC1 does not go to phase-separated nuclear droplets (Courchaine et al., Cell 184, 3612, 2021), and raises issues for the interpretation of the APEX protein interaction experiments with Tudor-only SMNDC1 of Figure 2. Moreover, the authors of this manuscript show in Fig.1k that the Tudor-only SMNDC1 does not phase-separate in vitro and, instead, the C-terminal unstructured region is necessary and sufficient for phase separation in vitro. That means that the phase separation observed by the authors in vitro is not necessarily related to the behavior of SMNDC1 in vivo. Finally, it is not clear from the data in the paper that SMNDC1 forms phase-separated nuclear droplets all on its own or is incorporated into droplets formed by some other protein(s), and this should be made clear in the text.

2. The resolution, or rather lack thereof, of the nuclear images in Figures 1 and 2 makes me wonder whether the authors are even looking at phase-separated droplets in vivo or, rather, simple occupancy of inter-chromatin spaces by the SMNDC1. Demonstration of such droplet formation in cells usually involves demonstrating diffusion of the protein and fusion of well separated droplets containing the droplets.

3. The number of proteins identified as in the vicinity of SMNDC1 by APEX is extraordinarily large (~3500), so large that it may actually be the majority of proteins in the nucleus. The APEX may simply be picking up most proteins present in inter-chromatin spaces, perhaps because of over-expression of the APAX-tagged protein, and it is not useful to simply picking up most of the proteome of the nucleus. That the overlaps with arginine-demethylated proteins, the SRSF7-APEX nuclear speckle proteins, and SMNDC1 co-IP proteins (Fig.2d,e,f) is large may simply reflect high overlaps with all nuclear proteins, all chromatin-excluded nuclear proteins or, given that the sensitivity of mass spectrometry may come into play, all nuclear proteins over a certain abundance threshold.

4. For the same reason, the GO enrichments of Fig.2c may not be meaningful. The enriched categories all relate to mRNA, rRNA, and ncRNA processing, ribosome biogenesis, and translation, and these are all large GO Biological process categories. If the GO process genes in the library that was used numbered 7683 (as in reference 39), and all were used, one would expect enrichment for everything in the nucleus, since expression was exclusively there. Moreover, if a high % of each GO category of nuclear proteins not in chromatin was identified at random, the best p values would be for GO categories with the largest numbers of proteins. Consistent with that, even though the average GO process category has 78 proteins, the enriched categories had, in order, 483, 296, 296, 313, 218, 413, 6334 (“gene expression”), 255 143, and 123 proteins.

5. That only 4 proteins (Fig.2g) were enriched in the APEX experiments for the full length SMNDC1 compared to Tudor-only SMNDC1 is consistent with the deletion construct lacking the IDRs going to the same places in the nucleus as full length and also inconsistent with the phase separation behavior observed in this manuscript in vitro and elsewhere in vivo (Courchaine et al., Cell 184, 3612, 2021). Not surprisingly, therefore, interacting nuclear proteins were also not depleted with truncated SMNDC1 (Suppl. Fig.2c).

6. Maybe I am not understanding something, but I find it striking that IC50 value of compound 13 for SMNDC1, namely 2.6uM (Fig.4), is so wildly different from its kD in NMR of 1mM (line 261).

7. The phase separation behavior of SMN depends on its interactions with dimethylated arginine (Courchaine et al., Cell 184, 3612, 2021), and one would expect the same for SMNDC1 if, as in this paper, its Tudor domain goes to nuclear droplets. Therefore, one would expect that the compounds of this paper, which target the Tudor domain, would quickly change the distribution of SMNDC1. However, the authors treat cells with compound 1 for 12-16 hours (Fig.6a-c) and say that this is just short of the time to cause cell death. This affects the distribution of SMNDC1, but it also affects the distribution of SMN and a nuclear speckle protein. There is no way to know from these experiments that the effects on the cell of compound 1 are because it specifically targets SMNDC1. One cannot even tell whether the effects of compound 1 on the distribution of SMNDC1 itself after 16 hours are because it actually targets SMNDC1. All of this is related to the issue that we cannot tell how specific is the interaction between the compounds and SMNDC1. The authors show that their compounds can distinguish between SMN and SMNDC1, but not that they do not bind or affect any other proteins, even ones with Tudor domains. To figure that out, the authors would have to deplete SMNDC1 (e.g. with a degron) and then show that compound 1 has no effect.

8. The authors say that they tested specific effects of compound 1 by showing effects on splicing in Fig.6g of 3 transcripts known to be affected by SMNDC1. It is impossible to know how specific this is since an unspecified number of transcripts showed no effects, and we do not know either how

reproducible the effects are (no statistics) or what happened to transcripts not targeted by SMNDC1. One would probably need a comparison of genome-wide analyses of splicing in cells depleted of SMNDC1 and cells treated with compound 1. I should also add that the methods used in the splicing analysis are not specified in the Methods section.

9. It is hard to reconcile the results in Fig.6h, in which compound 1 significantly reduces the biotinylation by APEX-tagged SMNDC1 of only 126 proteins in a proteomics experiment, with the results in Fig.6j, in which compound 1 has a massive effect on biotinylation of almost all proteins in a western blot. As well, it is hard to understand Suppl Fig.5h in which compound 1 does not reduce any proteins from the vicinity of the Tudor domain of SMNDC1, because I would have assumed that the interactions of the Tudor-only construct would involve the Tudor domain interacting with dimethyl-arginines on target proteins (in fairness, compound 1 did not much reduce interactions of the Tudor domain in a western blot either in Fig.6j).

RESPONSE TO REFEREES

We would like to thank all referees for their insightful comments. In addressing these, we have been able to significantly strengthen the manuscript in this revised version. In particular, we added the following new data:

- We now redid most of the microscopy imaging studies on both endogenously tagged proteins and by immunofluorescence. Thereby we were able to significantly improve image quality, at the same time confirming all the major conclusions of our study.
- We added new data using fluorescence recovery after photobleaching (FRAP). These new results are in line with phase separation behavior of SMNDC1. Furthermore, they show immediate effects of compound treatment on protein mobility, which precede the changes in foci formation and subcellular protein localization at later timepoint.
- We performed comprehensive 9-point dose responses of all compounds, including new compounds to expand structure activity relationships. We expended the dose points tested and used newly purified proteins and profiled all compounds in parallel. While this resulted in shifts in IC50 values and specificity profiles for some compounds, the data confirm the submicromolar activity of the most potent compounds and the overall structure-activity relationships we described.
- We now included a negative control compound in cellular experiments and show that in line with its absence of activity in the biochemical assay, we also observe no changes to cellular SMNDC1 localization with this compound.
- We also conducted paired-end RNA-sequencing of compound-treated cells and show that similar splicing effects on enhanced intron retention and exon skipping as we had previously observed with SMNDC1 knock-down.

We hope that these new data along with the text changes we incorporated in the revised version will make the manuscript suitable for acceptance. Please find our point-by-point responses below.

REVIEWER COMMENTS

Reviewer #1 (Remarks to the Author):

The manuscript by Enders et al. aims to characterize phase separation properties of SMNDC1 and identify small molecules using an AlphaScreen to impair this phase-separation property. Furthermore, the authors present the structure of the SMNDC1-Tudor domain in complex of one of the identified inhibitors. In general, the presented study is of high interest and might be suitable for Nature Communications. Nevertheless, despite the impressive amount of work the manuscript at the current state fails to be entirely convincing as outlined by the different points listed below. In addition, the manuscript reads at the moment at instances still as a sum of different parts and not one coherent text, which the authors might also want to address.

We thank the referee for finding our study of high interest in principle fitting to Nature Communications. In this revised version, we have now added new data and also worked on the coherence of the text.

Major Points:

The readability of the introduction should be improved as currently the paragraph about the Tudor domains (lines 70–75) seems to be oddly placed and not well integrated with the rest of the text.

We have rewritten the introduction to include a section which emphasizes the role of Tudor domains in phase separation which hopefully improves the integration of this paragraph with the rest of the text.

As the authors analyzed the intrinsic disorder as well as used the AlphaFold model it looks like a miss that they did not connect these in a better way (Fig. 1A; lines 80-93). For example does the disorder predictor correctly indicated the helices found in the AlphaFold model and based on how many co-evolution constraints did the AlphaFold model base the helix:helix interaction on.

We have now added a section to the text that directly compares AlphaFold and disorder predictions, describing the consistent prediction of the Tudor domain structure an N-terminal alpha helices, whereas the C-terminal helix is only predicted by AlphaFold without additional experimental evidence.

We do not think that it is easily possible to directly answer to the question on co-evolution constraints the AlphaFold model based the helix:helix interaction on. For an estimate of the consistency of these predictions, we checked the best-ranked AlphaFold models for human SMNDC1 and its closest homologues. All of them show the helix:helix interaction with 6-7 contact residues and a predicted distance between the helices of around 7 Angstroms. Also, the predicted aligned error matrix shows that AlphaFold predicts a very low error for the interaction between the two helices (residues 2-25, and 30-52).

A better labeling of the AlphaFold model is also needed as at the moment it is not entirely clear where the N-terminal and C-terminal extensions are.

We have added labels to the N-terminus and C-terminus in the AlphaFold-predicted structure in Fig. 1a.

It might be beyond the scope of the current manuscript it would be highly informative to have some experimental data on the full-length protein such as CD spectra, to confirm the existence of the helices in the rather unstructured regions, as AlphaFold tends for some IDPs (e.g. Tau40) to indicate helical content, which experimentally could not be verified.

We agree with the referee that that the long C-terminal helix might be an AlphaFold artifact. We have not been able to obtain experimental evidence for the existence of this helix and we now explicitly state in the manuscript that no such evidence exists currently beyond the AlphaFold prediction. The findings of our manuscript are not impacted by whether this helix exists or not, and therefore we have not extensively investigated this.

The additional benefit of the different AlphaFold structures with the GFP inserted in the intron regions is not clear to this reviewer and they seem to overcomplicate the current figure one.

We agree with the referee and have now moved the predicted structures for SMNDC1 with the GFP-tag integration to the supplementary Fig. 1. Instead, we now indicate the positions of the GFP integrations on top of the wild-type AlphaFold prediction in Fig. 1a.

Figure 1 in vitro droplet formation: It was a bit unclear how the authors actually performed these assays, whereas the figure legend and the main text (lines 112–124) indicated the use of either PEG-8000 or RNA in the methods section (lines 635–643) the author only mention the usage of PEG-8000. It should be clarified how the experimental set-up was.

For clarity, we have now moved in vitro droplet data to Fig. 2. We investigated droplet formation in the presence and absence of RNA, and now added buffer conditions to the main text, and the usage of RNA to the methods section. We now also indicate that RNA was used to Fig. 2f (former Fig. 1k).

The authors should reconsider the set-up of figure 4 as it is hard to assess at the moment. Maybe putting compound 1 in the center enlarged and rearranging the different modifications around would make this figure more accessible.

We have updated Fig. 5 (previous Fig. 4) according to the new SAR data. We more or less kept the layout, with compound 1 enlarged in the center and different substituents by highlighted color-coding arranged around that one. To us, this appears the most intuitive and accessible representation after trying out several alternatives.

The structures of compound 13 bound to SMNDC1 in Figure 5a and 5b look slightly different, which is maybe due to showing the protons of the H₂N group in panel A. Furthermore, as panel B seems to be slightly rotated this should be indicated.

Figure 6a and 6b (former Figure 5a and 5b) as well as Supplementary Fig. 4c were re-done and the structures are now oriented in the same way. The amino protons in Figure 6a were removed to avoid confusion.

Figure 5C, in the NMR spectrum are about four additional peaks experiencing large chemical shift changes that were not annotated. A detailed reasoning for the resonances chosen to be labeled would likely resolve this.

All peaks in Figure 6C (former 5C) with reasonably large chemical shift perturbation are now indicated. Peak N113 was formerly unassigned but could be assigned after re-evaluation of the datasets. Therefore, the bar plot in Figure 6a was also changed accordingly.

Figure 5D, it is definitely an important panel for the detection of the intermolecular NOEs. In slight contrast to the shown panel, which indicates the unspecific assignment of the ring-protons of the aromatic ring protons, the list of the intermolecular NOEs indicates stereospecific assignment for the ring-protons.

The aromatic protons of Tyr and Phe were indeed not specifically assigned (only one signal was observed for H δ 1/ δ 2 and H ϵ 1/ ϵ 2, respectively) and included in HADDOCK as ambiguous distance restraints. Supplementary Table 3 was generated from the distances in Supplementary Figure 4a where only one proton was selected (the closest) to reduce complexity in the figure. Supplementary Table 3 was adjusted and the caption of Supplementary Figure 4a was updated.

Minor Points

Line 134 – abbreviations should be introduced, thus it is missing for IF.

We now spell out immunofluorescence (IF) at the first occurrence in the text.

Lines 149-151 – the sentence ending with ...which are often the same proteins. Remains unclear. Same proteins identified in different datasets or do the authors mean the same proteins bind symmetrical and asymmetrically methylations. Please clarify.

We have added a sentence to clarify that the same arginine sites can often alternatively be symmetrically or asymmetrically di-methylated.

Line 398 – *In vitro* should be in italic

We have changed the font of all occurrences of “*in vitro*” to italic.

Figure 2c legend: The details of the Enrichr analysis here and in the maintext (line 146) should rather be presented in the methods section.

We have moved the details for the Enrichr analysis details from main text and figure legend to a dedicated point in the methods section.

Line 420 – adj. should read adjusted

We have changed all instances of “adj.” to adjusted.

Lines 427-430 – The text describing panels c and d seems to have a couple of grammar issues

We have now rewritten this figure legend to correct grammar and use full sentences. Line 430 – should readhits marked in yellow....

Actually, the remaining specific hits had been marked in red in this panel. For clarity, we have now removed this panel and rather show full 9-point dose responses for all hit classes.

Suppl. Figure 4 – the orientation of Panel A in relation to Fig. 5 should be indicated. In Panel C some grey ticks are shown, which seems to be an artifact.

The orientation of Supplementary Fig. 4a in respect to Fig. 6 (former Fig. 5) is now indicated and figure caption was updated. Supplementary Fig. 4c was re-done to answer an additional comment (see below).

- Suppl. Figure 5k – The RNAbindRplus Score could possibly related to the AlphaFold structural model

We did not observe an obvious correlations between RNA binding score and the AlphaFold structure. The C-terminal alpha helix is predicted at residues ~140-186, while the block of high RNAbindRPlus scores stretches from ~177-203 with some more peaks until the end of the protein (amino acid 238). Although a single helix has been reported to mediate RNA-binding (Tan et al. Cell 1993), most of the time more

complex structures and especially IDRs mediate RNA-binding (Corley et al. Mol. Cell 2020) which fits to the unstructured region predicted by AlphaFold following the C-terminal helix.

Suppl. Table 2 – In some of the compound cells red triangles are shown, which are not discussed and are possibly an artifact

Indeed, this was an artifact from copying the table into the document. The problem has been solved in the current version of the manuscript.

Suppl. Table 4 – some values in the HADDOCK table need clarification and maybe discussion in the main text: why is the error of the HADDOCK score 0? Same for the z-score. Also the RMSDs of 0.6 ± 0.6 (overall) and especially 0.61 ± 0.788 (pairwise) with the error being larger than actual value look suspicious.

Due to the rigid body docking procedure and the very similar binding mode of compound **13** and sDMA (of which the PDB structure was chosen as template), the protein structures of SMNDC1-compound **13** complex are indeed very similar (as enforced by the rigid body docking), hence resulting in very low values for the overall and pairwise RMSD, in part lower than the error. An overlay of the lowest energy structures in the resulting cluster was added to Supplementary Fig. 4c to illustrate this and the figure caption was updated.

HADDOCK score and z-score are output values of the HADDOCK webserver to evaluate the generated clusters. The HADDOCK scoring function consists of various energy terms and the buried surface area while the z-score indicates the deviation of the cluster from the average. Only one cluster was generated during our run, therefore the z-score is 0. We removed the latter one from the table.

Line 613 – *E.coli* should be in Italic

We have changed this to italic.

Lines 618-623: 4000g should be 4000xg to be consistent with other parts of the methods (see e.g. lines 713–727)

We have now changed all centrifugation specifications to “xg”.

Lines 728/733/739 –H₂O should be H₂O

We have changed this to H₂O (subscript).

Lines 773–794 – As SDS PAGE is a standard method and the others did not seem to use any particular special approaches the composition of the Lämmli buffer might not be necessary to provide. Rather important details like the used voltage, run length and type of the used Gels are on the other hand not provided.

We have added details for percentage of gels, voltage, run

length. Line 876 – should likely read....saturation, an additional

point... Yes, we have changed this accordingly.

In general the references need some clean-up as several different styles were used...

As requested in the guide to authors, we have used standard "Nature" reference style, with all authors listed unless there are six or more authors, in which case only the first author is given, followed by 'et al.' We double checked but could not find any inconsistencies, please point us to specific instances in case there are any problems remaining.

Reviewer #2 (Remarks to the Author):

This is an interesting paper linking Tudor domain interactions with dimethyllysine, and intrinsically disordered protein (IDP) interactions with RNA binding as two key molecular interactions that drive liquid-liquid phase separation (biomolecular condensates; membrane-less organelles). The study reports new chemical compounds that bind to the aromatic cage of SMNDC1 Tudor domain and appear to inhibit the interaction with sDMA, with consequent mildly reduced number of GFP-positive signal in phase separated nuclear speckles. However, this compound is of low affinity and therefore difficult to use for drawing firm conclusions. Overall the manuscript would benefit from the use of additional controls (such as suggested below) and better quantification of selected binding affinities by a complementary method other than just AlphaScreen assay (ideally with recombinant protein), and more consistent use of construct/protein nomenclature. Specific suggestions for improvements are listed below.

We thank the referee for this assessment, and have now further improved the manuscript in this revised version to include additional controls.

1. Fig 1. Placement of GFPs within 3D structure is hard to see within the alpha fold simulated structures. Instead annotation of where the GFP insertions are on the WT alphafold prediction would be best, and then use a schematic to depict the fusion proteins instead of complicated alpha fold structures. Authors can include the current AF structures in supplemental materials for readers who wish to see.

We thank the referee for this suggestion and have now updated Fig. 1 accordingly and moved the intron-tag AlphaFold structures to the supplement.

2. Intron2-3 GFP insertion appears to disrupt the small (predicted) alpha helix, but this is the construct used throughout most of the paper. Have the authors compared several of the phenotypes of Intron2-3 with the other insertions to show that 2-3 is really representative and not a fluke?

We confirmed the effect of the inhibitor on SMNDC1 distribution as the most important phenotype with all other insertions (Fig. 7a) and by immunofluorescence in WT cells (Supplementary Fig. 5f).

3. Nomenclature: 'SMNDC-GFP Intron2-3' is exemplified in Fig 1C, but then all remaining nomenclature is 'SMNDC1-GFP'. Are all subsequent expts done with the Intron 2-3 insertion? This ambiguity should be fixed.

Indeed, we had used intron 2-3 when the label just said SMNDC1-GFP. We have now changed the labeling to make it clear which intron-tag was used.

4. Nomenclature: Is 'truncated SMNDC1-APEX2' (in text) the same as SMNDC1-TudorDom-APEX (Fig2b)? Nomenclature should be consistent or each term better defined.

For the proximity labeling, we now consistently refer to the Tudor domain only construct as APEX2-SMNDC1^{TD} and to the full-length construct as APEX2-

SMNDC1^{FL}. We also included the constructs as Suppl. Fig. 2a which we hope will help to explain the different constructs used and how they are named throughout the paper.

5. Nomenclature: Does the truncated protein correspond to one of the constructs indicated in Fig1a? It would be good to map this construct (and others named, including that used for NMR) onto Fig 1A and include the protein names in Fig 1A.

We have introduced a new figure (Suppl. Fig. 2a) to clarify which APEX2-fusion proteins were used and how they are structured. In principle, the APEX2-SMNDC1^{TD} includes the same amino acid residues of SMNDC1 as construct “3” in Fig. 1A. Since construct “3” is a GFP-fusion while APEX2-SMNDC1^{TD} is fused to APEX2, we felt it might cause more confusion if we also include the APEX2-fusions in Fig. 1A.

6. The observation that SMNDC1 itself was enriched in full length over truncated SMNDC1-APEX2 could be due to fewer lysine-containing sequences available to be biotinylated in Tudor alone.

The Western Blot results show that also interaction to endogenous SMNDC1 is reduced. We have also confirmed this result with new western blots showing fusion-proteins (Suppl. Fig. 6d). We have also added a sentence to the main text to make this clearer.

7. Pg 8. To conclude (suspect) that “the Tudor domain is responsible for a major part of SMNDC1’s specific interactions”, the better experiment would have been to use a construct with only the Tudor domain deleted. The authors had concluded that the C-term (post-Tudor sequences) drives droplet/speckle formation/localization, and the Tudor alone was less restricted to speckles. It does not follow necessarily that the Tudor harbors all the specificity. Only a small subset of the full length APEX2 interactome were SDMA proteins. This logic does not follow.

We agree with the referee that both the tudor domain and the other parts of SMNDC1 including the C-terminus contribute to the observed interactions. We have now toned down this statement to state that a subset rather than a major part of interactions is driven by the Tudor domain.

We have also attempted to follow the referee’s suggestion to perform APEX2-interactome mapping with a SMNDC1 Tudor domain point mutant. However, we observe that also in this construct likely additional changes to protein stability and/or localization contribute to induce more widespread interactome changes.

8. Pg 8: “with IC₅₀ values of 3 to 20 μM” should be qualified as “in the range of ...” or ‘low micromolar’, as IC₅₀ values from a 4pt titration are not at all accurate.

We have now re-measured all IC₅₀ values of compounds mentioned in the paper with 9pt titrations and updated IC₅₀ values accordingly.

9. 50uM is a high concentration for cellular studies. At this concentration it is likely that other unintended proteins may also be substantially affected. Indeed, the authors state that at longer time points cells start to die. Can the authors be certain that the molecular events they report, such as loss of SMNDC1 from the nucleus is

due solely to inhibition of Tudor domain-sDMA binding? As a control for off target effects, the authors could use one of the inactive, but chemically similar compounds (including similar predicted cell permeability properties), from the SAR series. The inactive compound should not show any changes in protein localization at 50uM.

We agree with the referee that assessing ruling out off target effects is always challenging in cellular assays. That cells die in the long-term not necessarily confirms off target effects, as SMNDC1 is an essential gene and long-term knock-down of SMNDC1 is toxic to cells as well. We have now tested compound 9, which lacks the 2-pyridyl crucial for the binding to SMNDC1. Indeed, we did not see changes in protein localization at a concentration of 50 μ M. Furthermore, we have tested a low-dose, long-term treatment which also leads to comparable effects to high-dose overnight treatment in terms of alternative splicing. Finally, we have also done FRAP experiments to assess more immediate compound-induced changes to SMNDC1.

10. In the discussion, the authors state: Likely these RNA-mediated interactors together with arginine methyl interactions mediated by the SMNDC1 Tudor domain constitute the multivalent binding platform that is a typical prerequisite in the formation of biomolecular condensates. Given the confounding issues of using very high compound concentrations, further support for this notion could be provided by mutation of key aromatic cage residues in the Tudor domain. If the mutations phenocopy inhibitor treatment, this would also help validate compound 1 as a functional Tudor domain inhibitor in cells.

In the limited time for revisions, we were not able to introduce site-specific mutations or a knock-out combined with overexpression of SMNDC1 with a mutated Tudor domain. However, we have now included experiments with knock-down of SMNDC1 and show that these phenocopy the effects of a small-molecule Tudor domain inhibition, with no further changes resulting from inhibitor treatment of knock-down cells. While not ruling out off target effects, these data are consistent with SMNDC1-specificity of the observed effects.

Reviewer #3 (Remarks to the Author):

The manuscript “Pharmacological perturbation of the phase-separating protein SMNDC1” establishes that SMNDC1 partially co-localizes into nuclear speckles, which has been also proved by analysis of its ligands. SMNDC1 is the name of the gene encoding SPF30 protein (<https://www.uniprot.org/uniprotkb/O75940/entry>), a splicing factor with roles in spliceosome assembly and function. We suggest that SPF30 be used throughout the manuscript to refer to this protein to avoid confusion with SMN (Survival Motor Neuron protein).

To reduce confusion due to the difference between gene and protein name we have decided to only use SMNDC1. We now specifically state the usage of SMNDC1 also for the protein in the methods section. We hope SMNDC1 is distinguishable enough from SMN with the explanation how these two share a similar Tudor domain.

We are concerned that the low quality of the images that should define the compartment to which SPF30 localizes prevents a clear understanding if these are standard nuclear speckles, a subset, or a different but overlapping compartment. To answer these questions, immunofluorescent analysis of fixed cells by confocal, high resolution wide-field fluorescent light microscopy, or even STED should be used – the only microscopy documented in the manuscript is live imaging, so it is not clear how this analysis has been done.

We apologize for the low quality of the images in the pdf version of the manuscript generated through the online submission tool. We hope that this has been fixed now by uploading high resolution figures separately.

We had reported immuno-fluorescence staining also in the previous version of the manuscript, and have now also taken new confocal, high-resolution images of both live and fixed cells with an LSM980 microscope and analyzed co-localization (Fig. 1g-h, Suppl. Fig. 1e).

There are also several instances where methodology is not sufficiently described to allow reproduction.

We have improved the methods section with more details. Please let us know details in case you still find important information missing.

The co-localization of SPF30 protein into nuclear speckles and its overall function in splicing may be specifically disrupted by a new chemical inhibitor. Although tudor domains are increasingly recognized for their role in biomolecular condensation, chemical inhibitors of specific tudor domains/tudor domain containing proteins have so far been lacking. This brings impact to the present manuscript. However, inhibition of SPF30’s role in splicing – its best characterized function – has not been adequately tested. Thus, the characterization of the effects of disrupting SMNDC1 function must be improved. Below, we make constructive suggestions for improving the manuscript and point out a number of specific weaknesses that prevent publication of the manuscript in its current form.

We now included new data on paired-end RNA-seq with compound treatment, and observe comparable changes on intron retention and exon skipping as we have

previously reported for SMNDC1 knock-down (Fig. 7jk). We have further expanded the panel of transcripts for which we have PCR-based validation, and there in addition to compound-induced changes we also show that compounds typically do not induce any further changes when used with SMNDC1 knock-down cells. (Fig. 7l, Supplementary Fig. 6).

Detailed comments:

1/ Figure 1 is hard to understand because it is too big and complicated and there are multiple conclusions. We think this should be split in half (new fig 1 and 2), perhaps separating the *in vivo* from the *in vitro* experiments. This would enable the authors to describe more rigorously the GFP-truncated proteins in the text as well as in the figure legend.

Thank you for this suggestion. We have now split Figure 1 in two, separating SMNDC1's localization and co-localization with SRRM2 (in new Fig. 1) from the phase-separation analyses both *in vitro* and in cells (in new Fig. 2).

Line 95 – The authors claim that all structural elements are predicted normally. However, the protein with disrupted Tudor domain is obviously expressed less than other protein versions (Suppl. Fig. 1a). Please reflect this evidence in the main text, ideally by quantifying the tagged versus endogenous protein in 3 replicates.

We have now quantified tagged versus endogenous protein in 4 replicates (Supplementary Fig. 1bc). We have also added a sentence mentioning the lower expression levels of GFP intron tag 3-4.

Regarding the Suppl. Fig. 1b, could the authors show a different field of cells overexpressing CLK1? It is difficult to interpret the staining relative to cell morphologies.

We have added a different field of view of CLK1 overexpressing cells (now Supplementary Fig. 1d).

Fig. 1d – The colocalization of SMNDC1 with Mitotic Interchromatin Granules must be quantified, as well as the “partial” co-localization of SC-35 and SMNDC1 in interphase. Was this partial co-localization also observed in the case of Mitotic Interchromatin Granules, or did they co-localize completely during mitosis?

We have used new high-resolution, confocal z-stack images to quantify co-localization both in interphase as well as in Mitotic Interchromatin Granules (Fig. 1g-h, Supplementary Fig. 1e-f). Pearson correlation values were on average lower in mitotic cells compared to interphase cells.

“Mitotic Interchromatin Granules” is only used once, so we suggest getting rid of the acronym MIG.

We have now removed the acronym.

Fig 1f – Hexanediol is really a dubious way of implicating LLPS as the underlying process, due to other effects of hexanediol on cells. For example, it inhibits kinases

and phosphatases (<https://pubmed.ncbi.nlm.nih.gov/33814344/>)! A much better way of addressing reversibility in this system would be FRAP, which would also be more easily quantifiable with more interpretable parameters like % recovery rather than intensity per nucleus or spots/nucleus, which are more ambiguous.

We have now set-up FRAP experiments with our GFP- and RFP-tagged cell line (Fig. 2h, and plus inhibitor Fig. 7g) which shows the fast recovery of both SMNDC1 and SRRM2 as expected for phase-separated proteins.

Fig. 1g – The overexposed GFP images are not explained. It is preferable to have the same microscope setting in all conditions.

We have replaced the images with non-overexposed images.

Fig. 1k – Please add information about the presence of RNA into the figure for better understanding.

We have added the information about the presence of RNA (now Fig. 2f).

2/Figure 2:

Fig. 2b – The image quality is very low. It appears that the APEX2 construct fills the entire nucleus, as does the streptavidin signal. This could account for the large number of proteins identified. Could you show the cell boundary to provide us with better information about the expressed proteins in the cytoplasm?

We agree with the referee that the broad localization of the streptavidin signal might be a contributing factor in the number of proteins identified, although the signal is not entirely homogenous. We have not performed a membrane stain in these experiments, therefore do not have a marker for cell boundaries.

In initial experiments, we had attempted to minimize labelling time aiming as specific labeling as possible. Unfortunately, we could not get a better specificity, also in a new attempt for this experiment in the revision process.

Fig. 2d – Could you please clarify your statement on sDMA proteins on lines 147-149, since there is only an overlap of 67 proteins?

We have added the numbers of proteins in parentheses. The total number of known sDMA proteins is rather low (87 proteins), and most of these were identified. We feel “most” proteins as a term to describe the high fraction of known sDMA proteins identified in our proximity labeling experiments is accurate.

Fig. 2h – SRSF7-APEX2 served as a control nuclear speckle protein. We think this should be removed, because there is no negative control for the SRSF7-APEX experiment. Also, the information provided is not directly related to the main argument.

While it is true that the published SRSF7-APEX2 dataset is from a different cell system, we still would like to keep these data in to emphasize the overlap to known speckle interactors. We view this overlap plot as provided supporting information in

the same way we would use gene set/GO term enrichment analysis for transcriptomics data.

In the text related to Figure 2, the specificity of SMNDC1 towards sDMA and aDMA, which was published in Tripsianes, Nature Structural & Molecular Biology, 2011, should be mentioned because it is important to the interpretation.

We have added this reference once more also to this sentence.

The main issue of the manuscript is that information on SMNDC1 ligands (Figure 2) is not fluently connected to perturbation of SMNDC1 function. Thus, the motivation for perturbing SMNDC1 function is disconnected from the results so far.

We believe that chemical tools can play important roles to dissect protein function in cellular systems, that are also translatable to tissues that are hard to engineer with genetic methods. We explain this motivation for developing pharmacological agent to perturb SMNDC1 function also in the discussion.

3/ Figure 3:

Fig. 3f-l – Please include the IC₅₀ for SMNDC1 and SMN1 in the figures for better orientation.

We have updated this figure (now Fig 4f-k) and included the IC₅₀ values according to the new AlphaScreen data.

Please explain the abbreviation SAR on line 204.

We have added the explanation for the abbreviation of structure-activity relationship (SAR).

4/ Would it be possible to display a visualization of your statement of “the β 2-strand” from lines 259-260 in Figure 5? The position of this strand is not reflected in the figure. The authors should also elaborate on the significant chemical shifts in Figure 5 for a broader audience that could be unfamiliar with NMR.

We changed Fig. 6 (former Fig. 5) accordingly. We have included a sentence to explain the chemical shifts.

5/ Please refer to the compounds with a name like “compound 1” or even C1 for short. Please do not refer to the compound in the sentence as “1”. This is far too confusing.

We apologize for the confusion arising from only bolded numbers for compounds as is sometimes done in the medicinal chemistry literature. We have now replaced all instances where just the number was used with “compound 1” or “inhibitor 1”.

6/ Figure 6:

This figure is not well laid out. It could be split into two figures or certain panels could be moved to the supplement. The fonts are too small to see even allowing the figure to take up the whole page.

We have now rearranged this figure to show all data for the cellular characterization of compound effects split between Fig. 7 and Supplementary Figs. 5-6.

The canonical function for SPF30 is in splicing. The data asking if the inhibitor shows predicated changes in splicing are completely unacceptable. We think this experiment should be done globally and the chemical inhibition should be compared to and SPF30 knockdown or depletion by degron. High quality, low cycle number, quantified RT-PCR could be an option in the absence of global data. Another option would be to add the compounds to an in vitro splicing reaction and show dose-dependent effects on splicing. Instead, we are shown completely saturated images of bands for which there are no proper positive or negative controls and the purported results are not believable (panel g). It is NOT convincing that splicing is inhibited.

We have now added data of a genome-wide analysis of splicing by RNA-sequencing after compound 1 treatment. This shows that SMNDC1 inhibition with compound 1 has similar effects on splicing as a knock-down of SMNDC1, namely an increased retention of introns and a skipping of exons (Fig. 7j-k). A number of the genes differentially spliced both by knock-down and compound 1 inhibition could be confirmed by PCR (Fig. 7l, Supplementary Fig. 6f). In these figures we also provide direct comparison to SMNDC1 knock-down and analyze for compound effects in SMNDC1 knock-down cells.

The validity of the statistical analysis of the graphed data in this figure (panels b, c, e, and f) is questionable. Various parameters are analyzed by ratio paired t-test from live cell imaging. What are the numbers of cells and numbers of replicates considered?

Ratio paired t-tests were used since the different intron-tag cell-lines show different absolute SMNDC1-GFP intensities.

The quantifications for Fig. 7b and c were now replaced with an experiment including the negative control compound 9. These experiments were done in triplicate. Per replicate we analyzed between 2868 and 6711 cells. In the main figure, we have also replaced panels e and f by an experiment including negative control compound 9. Again, these were measured in triplicate and numbers of cells varied between 2868 and 5622.

In the experiments with the different SRRM2-RFP clones (now Supplementary Fig. 5d-e) cell numbers ranged from 737 to 4348.

Fig. 6a – Please show DAPI staining to indicate the limits of the nuclei and to show if cells are intact and/or in mitosis.

Since these were live cells, Hoechst was used instead of DAPI. This panel was now replaced with an experiment also containing images on the negative control compound 9, and the respective Hoechst staining images can be found in Supplementary Fig. 5a.

Fig. 6h&i – The volcano plot is very confusing. We do not understand the expectations regarding the volcano plot. If the compound is inhibiting binding to DMA ligands, why not display an analysis of the DMA ligands indicated in Fig 2?

We believe this is what we did in Fig. 6i, now Fig. 7h. We have marked all known sDMA-proteins found in the proximity analysis showing that these are preferentially lost upon treatment with the inhibitor.

Why not do a co-IP from treated versus control cells and ask if the amount of DMA-modified proteins that co-IP change? An analysis of annotated hits, such as STRING analysis, should be considered.

We think that proximity labeling is a better method to capture the interactions of SMNDC1 in the context of a phase-separated compartment like nuclear speckles. We have run an Enrichr analysis of the 124 hits in Figure 7h. It shows a very similar result as the analysis of proteins enriched in SMNDC1^{FL}-APEX2 over SMNDC1^{TD}-APEX2 (Fig. 3d). Terms that we consider typical for expected interactors of SMNDC1 like mRNA processing and splicing are the most enriched. So these interactors relevant for SMNDC1's canonical function are lost upon inhibitor treatment.

6/ Supplementary Figure 3:

Fig. 3b-c The authors should explain how the titration unit of protein-peptide, which is in 'nM', relate to the titration unit of beads, which is in 'pM/ml'. The relationship between these two titration experiments is not clear.

There is not necessarily a relationship between the two titration experiments. Unit of the beads is in pg/ml, as they are provided by PerkinElmer. At a final assay volume of 25 µl per well and 5 pg/ml concentration, this means 125 ng of each bead was used per well.

There is a theoretical binding capacity provided by PerkinElmer, but this very much depends on the exact proteins/ peptides used. For the Streptavidin-beads interacting with the biotinylated peptides this binding capacity is 30 nM at a bead concentration of 20 µg/ml. For the Nickel chelate beads used to bind the HisTag-Tudor domain, the binding capacity is 300nM- 1µM. We have added the protein and peptide concentrations used in the bead titration experiment (50 nM peptide, 75 nM protein).

7/ Conclusions regarding cell death based on Supplementary Figure 5:
Lines 291-293 say that “the majority of cells was not apoptotic or necrotic (necrotic refers to a tissue – this word should be removed and possibly be replaced with the descriptor that is more relevant to the authors) as shown by Annexin V or Propidium Iodide staining (Suppl 5a).”

We have changed necrotic to “undergoing other forms of cell death”.

We disagree with the interpretation. Based on the error bars, the differences shown in Suppl. Fig. 5a are significant. The plots seem to indicate that compound 1 leads to 2-3 times more AnnexinV or PI staining, respectively.

We agree that compound 1 leads to increased AnnexinV or PI staining. Our statement was “majority of cells was not...”. We have changed the sentence to clarify that the percentage of AnnexinV and Propidium Iodide (PI) positive cells was elevated.

How a snapshot in the staining experiment relates to the overall number of cells that are dying is not obvious. If the authors wish to monitor the number of living cells in the culture, perhaps they should use mitotracker or MTT to identify and analyze remaining living cells in image data.

We have used this timepoint (16h after start of the treatment) also used for the other read-outs to show that while there are already changes in the distribution of SMNDC1 and SRRM2, these changes precede cell death. Total cell numbers analyzed were 5944 in the DMSO condition and 4264 in the compound treated condition.

8/ Discussion

The pharmacological relevance of your study, the significance of obtaining SMNDC1 interactome, the role of SMNDC1 in condensation should be more rigorously discussed. More recent publications should be included and include more recent publications.

We have now expanded the discussion to more comprehensively cover the points mentioned.

Reviewer #4 (Remarks to the Author):

SMNDC1 (SPF30) is a component of the human spliceosome that interacts with dimethyl-arginines of Sm proteins of spliceosomal snRNPs. The authors describe the identification of compounds that bind to the dimethyl-arginine-binding pocket of SMNDC1 and, in some cases, distinguish between the pockets of SMNDC1 and the closely related protein SMN. Such compounds could be important for the future investigation of the importance of SMNDC1 for gene regulation and the determination of cellular identity. The authors also provide evidence that SMNDC1 phase separates and occupies nuclear bodies that partially overlap with nuclear speckles known to contain splicing components, as well as evidence that their compounds might disrupt SMNDC1 function. While I agree with the authors that the identification of potential small molecule inhibitors of SMNDC1 might well be important, I think that the data they provide about the normal behavior of SMNDC1, as well as the ability of their compounds to disrupt its function, has numerous flaws, as described below, that should preclude the publication of their manuscript in its present form.

1. In Fig. 2b the authors look at full length SMNDC1, as well as SMNDC1 containing only its Tudor domain, both fused to APEX, and examine SMNDC1 localization, as well as biotinylation by binding to streptavidin. As the authors point out, part of the Tudor-only construct is in the cytosol, presumably because it lacks its NLS. However, the large portion in the nucleus appears to be in the same phase-separated regions as the full length protein. This is a contradiction with the literature, where Tudor-only SMNDC1 does not go to phase-separated nuclear droplets (Courchaine et al., Cell 184, 3612, 2021), and raises issues for the interpretation of the APEX protein interaction experiments with Tudor-only SMNDC1 of Figure 2.

It is true that APEX2-SMNDC1^{TD} seems to avoid DNA-dense regions like APEX2-SMNDC1^{FL} does, but the resolution of these images and the fact that biotinylated proteins might change localization during the labeling duration make it difficult to assess APEX2-SMNDC1^{TD} localization to phase separated nuclear droplets solely based on these stainings. The proteins enriched in APEX2-SMNDC1^{FL} over APEX2-SMNDC1^{TD} overlap with the proximity interactome of SRSF7, a known nuclear speckle protein, and are enriched for mRNA processing and splicing factors. We therefore conclude that only APEX2-SMNDC1^{FL} (and SMNDC1^{FL}) localize to phase-separated nuclear droplets.

Moreover, the authors of this manuscript show in Fig.1k that the Tudor-only SMNDC1 does not phase-separate *in vitro* and, instead, the C-terminal unstructured region is necessary and sufficient for phase separation *in vitro*. That means that the phase separation observed by the authors *in vitro* is not necessarily related to the behavior of SMNDC1 *in vivo*.

We agree with the statement in so far as the *in vitro* droplet system is just a model which does not necessarily reflect the situation *in vivo*. Nevertheless, *in vitro* droplet experiments are widely used to study phase separation behavior of proteins in a controlled environment. We also show phase separation behavior in cells with 1,6-Hexanediol and we have now added FRAP data as well (Fig. 2h).

Finally, it is not clear from the data in the paper that SMNDC1 forms phase-separated nuclear droplets all on its own or is incorporated into droplets formed by some other protein(s), and this should be made clear in the text.

We have now added a point to the discussion stating that behavior in cells may likely be dependent the presence of other proteins.

2. The resolution, or rather lack thereof, of the nuclear images in Figures 1 and 2 makes me wonder whether the authors are even looking at phase-separated droplets in vivo or, rather, simple occupancy of inter-chromatin spaces by the SMNDC1. Demonstration of such droplet formation in cells usually involves demonstrating diffusion of the protein and fusion of well separated droplets containing the droplets.

We apologize for the low quality of the images in the pdf version of the manuscript generated through the online submission tool. We hope that this has been fixed now, and we are also uploading a separate pdf-file for the figures.

We have also taken new confocal, high-resolution images of both live and fixed cells with an LSM980 microscope and analyzed co-localization with the established marker for the phase-separated compartment of nuclear speckles, SRRM2 (Xu et al. Nucleic Acids Research 2022) (Fig. 1g-h, Supplementary Fig. 1e). Furthermore, we have established FRAP experiments which show the mobility and diffusion of SMNDC1 (and SRRM2) protein within the nucleus and nuclear speckles.

3. The number of proteins identified as in the vicinity of SMNDC1 by APEX is extraordinarily large (~3500), so large that it may actually be the majority of proteins in the nucleus. The APEX may simply be picking up most proteins present in inter-chromatin spaces, perhaps because of over-expression of the APAX-tagged protein, and it is not useful to simply picking up most of the proteome of the nucleus. That the overlaps with arginine-demethylated proteins, the SRSF7-APEX nuclear speckle proteins, and SMNDC1 co-IP proteins (Fig.2d,e,f) is large may simply reflect high overlaps with all nuclear proteins, all chromatin-excluded nuclear proteins or, given that the sensitivity of mass spectrometry may come into play, all nuclear proteins over a certain abundance threshold.

We have now run the same analyses only with the proteins enriched in APEX2-SMNDC1^{FL}- over APEX2-SMNDC1^{TD}-, which reduces the number of proteins which we consider relevant and specific interactors APEX2-SMNDC1^{FL}- to 750 (Fig. 3c-f). The results for Enrichr analysis and overlaps in principle stay the same, so we hope that we could address the concerns in terms of the large number of interactors.

4. For the same reason, the GO enrichments of Fig.2c may not be meaningful. The enriched categories all relate to mRNA, rRNA, and ncRNA processing, ribosome biogenesis, and translation, and these are all large GO Biological process categories. If the GO process genes in the library that was used numbered 7683 (as in reference 39), and all were used, one would expect enrichment for everything in the nucleus, since expression was exclusively there. Moreover, if a high % of each GO category of nuclear proteins not in chromatin was identified at random, the best p values would be for GO categories with the largest numbers of proteins. Consistent with that, even though the average GO process category has 78 proteins, the

enriched categories had, in order, 483, 296, 296, 313, 218, 413, 6334 (“gene expression”), 255 143, and 123 proteins.

We hope that with defining a subset of proteins enriched in APEX2-SMNDC1^{FL}- over APEX2-SMNDC1^{TD}- we could solve this problem (see answer above).

5. That only 4 proteins (Fig.2g) were enriched in the APEX experiments for the full length SMNDC1 compared to Tudor-only SMNDC1 is consistent with the deletion construct lacking the IDRs going to the same places in the nucleus as full length and also inconsistent with the phase separation behavior observed in this manuscript in vitro and elsewhere in vivo (Courchaine et al., Cell 184, 3612, 2021). Not surprisingly, therefore, interacting nuclear proteins were also not depleted with truncated SMNDC1 (Suppl. Fig.2c).

In Supplementary Figure 2c we show that nuclear proteins are clearly enriched in APEX2-SMNDC1^{FL}- over APEX2-SMNDC1^{TD}. This can also be confirmed in an Enrichr analysis GO cellular component 2021 of the 750 proteins enriched in APEX2-SMNDC1^{FL} over APEX2-SMNDC1^{TD}:

6. Maybe I am not understanding something, but I find it striking that IC50 value of compound 13 for SMNDC1, namely 2.6uM (Fig.4), is so wildly different from its kD in NMR of 1mM (line 261).

This difference is likely related to low solubility of the compounds. While concentrations of **13** in solution were estimated by NMR peak intensity, a linear correlation between signal intensity and concentration is only observed up to 1 mM concentration, indicating aggregation of the compound. The AlphaScreen assay is done at lower concentrations, and the signal also depends on the presence of multiple proteins immobilized to a given nanoparticle, thus increasing the apparent IC50 measured. As a result the AlphaScreen experiment it is not or less affected by the limited solubility of the compound. The NMR titrations were important to estimate the binding isotherm and ensure that the complex is in saturation for the structural

analysis. Therefore, an additional point with higher compound concentration (approx.

8mM) was recorded (see figure below).

The solubility issues do not allow an accurate K_d determination and the statement on the apparent K_d was therefore removed from the main text.

7. The phase separation behavior of SMN depends on its interactions with dimethylated arginine (Courchaine et al., Cell 184, 3612, 2021), and one would expect the same for SMNDC1 if, as in this paper, its Tudor domain goes to nuclear droplets. Therefore, one would expect that the compounds of this paper, which target the Tudor domain, would quickly change the distribution of SMNDC1.

Indeed, one would expect some immediate changes upon inhibition of the Tudor domain. Employing the newly established FRAP, we can now show that treatment with the inhibitor does immediately (after 15-45 min) reduce fluorescence recovery after photobleaching (Fig. 7i). We therefore speculate that mobility and incorporation of SMNDC1 within nuclear speckles is reduced, eventually leading to the changes in distribution observed at later time-points (12h).

However, the authors treat cells with compound 1 for 12-16 hours (Fig.6a-c) and say that this is just short of the time to cause cell death. This affects the distribution of SMNDC1, but it also affects the distribution of SMN and a nuclear speckle protein. There is no way to know from these experiments that the effects on the cell of compound 1 are because it specifically targets SMNDC1. One cannot even tell whether the effects of compound 1 on the distribution of SMNDC1 itself after 16 hours are because it actually targets SMNDC1. All of this is related to the issue that we cannot tell how specific is the interaction between the compounds and SMNDC1. The authors show that their compounds can distinguish between SMN and SMNDC1, but not that they do not bind or affect any other proteins, even ones with Tudor domains. To figure that out, the authors would have to deplete SMNDC1 (e.g.

with a degron) and then show that compound 1 has no effect.

We have now added data on SMNDC1 knock-down and a combination of compound 1 treatment with SMNDC1 knock-down (Fig. 7f). We observe that the knock-down of SMNDC1 shows very similar effects as the inhibitor treatment. Furthermore, a combination of knock-down and inhibitor does not increase effect size, consistent with both conditions acting on the same target.

8. The authors say that they tested specific effects of compound 1 by showing effects on splicing in Fig.6g of 3 transcripts known to be affected by SMNDC1. It is

impossible to know how specific this is since an unspecified number of transcripts showed no effects, and we do not know either how reproducible the effects are (no statistics) or what happened to transcripts not targeted by SMNDC1. One would probably need a comparison of genome-wide analyses of splicing in cells depleted of SMNDC1 and cells treated with compound 1.

We have now added data of a genome-wide analysis of splicing by RNA-sequencing after compound 1 treatment. This shows that SMNDC1 inhibition with compound 1 has similar effects on splicing as a knock-down of SMNDC1, namely an increased retention of introns and a skipping of exons. A number of the genes differentially spliced both by knock-down and compound 1 inhibition could be confirmed by PCR (Fig. 7l, Supplementary Fig. 6f).

I should also add that the methods used in the splicing analysis are not specified in the Methods section.

We have added a methods part on the splicing PCRs.

9. It is hard to reconcile the results in Fig.6h, in which compound 1 significantly reduces the biotinylation by APEX-tagged SMNDC1 of only 126 proteins in a proteomics experiment, with the results in Fig.6j, in which compound 1 has a massive effect on biotinylation of almost all proteins in a western blot.

The mass-spectrometric data is normalized to total peptides. Indeed, we could see that less total peptides (=less interacting proteins) were detected in the samples treated with compound 1.

As well, it is hard to understand Suppl Fig.5h in which compound 1 does not reduce any proteins from the vicinity of the Tudor domain of SMNDC1, because I would have assumed that the interactions of the Tudor-only construct would involve the Tudor domain interacting with dimethyl-arginines on target proteins (in fairness, compound 1 did not much reduce interactions of the Tudor domain in a western blot either in Fig.6j).

In line with this comment by the referee, we cannot fully explain these data. A possible explanation could be that in this setting where due to the loss of the nuclear localization signal the Tudor domain is no longer correctly localized, also its interactions are mainly unspecific and not specific Tudor-domain driven ones.

REVIEWERS' COMMENTS

Reviewer #1 (Remarks to the Author):

The revised manuscript by Enders et al. is in my opinion greatly improved compared to the previous version and in particular the additional experimntal data is highly appreciated. The authors addressed all my concerns as well as the majority of the other reviewers points. I congratulate the author to a very nice piece of work, which in my view is suitable for publication in Nature Communications.

Reviewer #2 (Remarks to the Author):

The revised manuscript addresses my concerns. It is much improved.

Reviewer #3 (Remarks to the Author):

Overall, the authors improved the manuscript substantially. They re-phrased many statements and conclusions that seemed to be too strong based on their experimental outcomes. Moreover, they improved the imaging studies and performed additional experiments that supported the conclusions, such as FRAP, adding substantially to the rigor of this study. The discussion is now better readable and the motivation for the work is better explained.

As a minor point, we still feel the presentation of the GO ontology/Enrichr analysis is unprofessional and poorly described as well; we hope the authors could consider a different form of graphical display that adds more to our understanding of the dataset.

RESPONSE TO REFEREES

REVIEWER COMMENTS

Reviewer #1 (Remarks to the Author):

The revised manuscript by Enders et al. is in my opinion greatly improved compared to the previous version and in particular the additional experimental data is highly appreciated. The authors addressed all my concerns as well as the majority of the other reviewers points. I congratulate the author to a very nice piece of work, which in my view is suitable for publication in Nature Communications.

Thank you for this positive assessment and for recommending our manuscript for publication.

Reviewer #2 (Remarks to the Author):

The revised manuscript addresses my concerns. It is much improved.

Thank you for this positive assessment.

Reviewer #3 (Remarks to the Author):

Overall, the authors improved the manuscript substantially. They re-phrased many statements and conclusions that seemed to be too strong based on their experimental outcomes. Moreover, they improved the imaging studies and performed additional experiments that supported the conclusions, such as FRAP, adding substantially to the rigor of this study. The discussion is now better readable and the motivation for the work is better explained.

We are very grateful for the helpful suggestions during the review process and are happy that the reviewer appreciates our efforts to improve the manuscript.

As a minor point, we still feel the presentation of the GO ontology/Enrichr analysis is unprofessional and poorly described as well; we hope the authors could consider a different form of graphical display that adds more to our understanding of the dataset.

We have now replaced the presentation of the GO ontology/ Enrichr analysis which was a simple list of the top 10 terms sorted by their p-value with a plot showing adjusted p-value and odds ratio. Furthermore, we have expanded the description of this analysis in the methods section and provide the full table of the results for GO Biological Process 2021 in the source data. These new analyses confirm the strong enrichment of a group of terms associated with mRNA processing, splicing, and rRNA processing/ ribosome biogenesis that we had also identified in the previous version of this figure.

For consistency with these new analysis and the updated figure, we would also like to amend our response to points raised by reviewer #4 regarding these experiments and enrichment analyses in the original first submission of our manuscript:

Reviewer #4 (excerpt):

3. The number of proteins identified as in the vicinity of SMNDC1 by APEX is extraordinarily large (~3500), so large that it may actually be the majority of proteins in the nucleus. The APEX may simply be picking up most proteins present in inter-chromatin spaces, perhaps because of over-expression of the APAX-tagged protein, and it is not useful to simply picking up most of the proteome of the nucleus. That the overlaps with arginine-demethylated proteins, the SRSF7-APEX nuclear speckle proteins, and SMNDC1 co-IP proteins (Fig.2d,e,f) is large may simply reflect high overlaps with all nuclear proteins, all chromatin-excluded nuclear proteins or, given that the sensitivity of mass spectrometry may come into play, all nuclear proteins over a certain abundance threshold.

We have now run the same analyses only with the proteins enriched in APEX2-SMNDC1^{FL}- over APEX2-SMNDC1^{TD}-, which reduces the number of proteins which we consider relevant and specific interactors APEX2-SMNDC1^{FL}- to 750 (Fig. 3c-f). The results for Enrichr analysis and overlaps in principle stay the same, so we hope that we could address the concerns in terms of the large number of interactors.

4. For the same reason, the GO enrichments of Fig.2c may not be meaningful. The enriched categories all relate to mRNA, rRNA, and ncRNA processing, ribosome biogenesis, and translation, and these are all large GO Biological process categories. If the GO process genes in the library that was used numbered 7683 (as in reference 39), and all were used, one would expect enrichment for everything in the nucleus, since expression was exclusively there. Moreover, if a high % of each GO category of nuclear proteins not in chromatin was identified at random, the best p values would be for GO categories with the largest numbers of proteins. Consistent with that, even though the average GO process category has 78 proteins, the enriched categories had, in order, 483, 296, 296, 313, 218, 413, 6334 (“gene expression”), 255 143, and 123 proteins.

We hope that with defining a subset of proteins enriched in APEX2-SMNDC1^{FL}- over APEX2-SMNDC1^{TD}- we could solve this problem (see answer above).

New, extended discussion:

The lower number of proteins (750 compared to ~3500 before) makes a considerable difference in terms of the meaningfulness of our analyses. We further chose a different form of representation to highlight that there is a significant enrichment for certain terms in the list of proteins that is enriched in APEX2-SMNDC1^{FL} over APEX2-SMNDC1^{TD}. We now show not only p-values which indeed “automatically” become lower the larger the number of proteins analyzed, but also the odds ratio. The odds ratio for the highlighted terms is between 16.97 (“mRNA processing”) and 26.91 (rRNA processing). A large proportion (up to ~50%) of all proteins connected to these terms are found in our dataset, as can be seen in the “overlap” column of the Source Data table:

Term	Overlap
mRNA processing (GO:0006397)	108/300
mRNA splicing, via spliceosome (GO:0000398)	101/274
RNA splicing, via transesterification reactions with bulged adenosine as nucleophile (GO:0000377)	97/251
ribosome biogenesis (GO:0042254)	87/192
rRNA processing (GO:0006364)	83/173
rRNA metabolic process (GO:0016072)	78/162
ncRNA processing (GO:0034470)	84/201

Furthermore, we analyzed a set of all proteins/ genes with a known nuclear localization (6177 genes) and found a different pattern of enriched terms (see below, top 10 terms by p-value). Transcription-related terms are much more enriched compared to our dataset, showing that it is not merely a random nuclear set of proteins, but a specific set related to splicing and rRNA. While terms such as “mRNA processing” and “mRNA splicing” show up in both protein sets, an 8-fold reduction in the size of the analyzed set (6177 vs. 750) does not lead to a comparable reduction in the overlap. For these two terms, we still find an overlap of 108/300 and 101/274 in our data set, compared to 258/300 and 236/274 in the much bigger “all nuclear proteins” dataset.

Term	Overlap	P-value	Adjusted P-value	Odds Ratio
regulation of transcription by RNA polymerase II (GO:0006357)	1528/2206	0	0	6.37
regulation of transcription, DNA-templated (GO:0006355)	1569/2244	0	0	6.63
negative regulation of transcription, DNA-templated (GO:0045892)	670/948	2.95E-147	5.34E-144	5.93
positive regulation of transcription, DNA-templated (GO:0045893)	783/1183	8.52E-147	1.16E-143	4.87
positive regulation of transcription by RNA polymerase II (GO:0045944)	626/908	5.86E-129	6.35E-126	5.42
negative regulation of transcription by RNA polymerase II (GO:0000122)	485/684	2.25E-106	2.03E-103	5.83
mRNA processing (GO:0006397)	258/300	6.46E-89	5.01E-86	14.30
negative regulation of nucleic acid-templated transcription (GO:1903507)	348/464	4.39E-87	2.98E-84	7.05
mRNA splicing, via spliceosome (GO:0000398)	236/274	1.19E-81	7.15E-79	14.41
transcription by RNA polymerase II (GO:0006366)	262/320	1.60E-80	8.70E-78	10.51